# IS MULTITASK LEARNING ALL WE NEED IN CONTINUAL LEARNING?

## ABSTRACT

Continual Learning solutions often treat multitask learning as an upper-bound of what the learning process can achieve. This is a natural assumption, given that this objective directly addresses the catastrophic forgetting problem, which has been a central focus in early works. However, depending on the nature of the distributional shift in the data, the multi-task solution is not always optimal for the broader continual learning problem. In this work, we draw on principles from online learning to formalize the limitations of multitask objectives, especially when viewed through the lens of cumulative loss, which also serves as an indicator of forward transfer. We provide empirical evidence on when multi-task solutions are suboptimal, and argue that continual learning solutions should not and *do not* have to adhere to this assumption. Moreover, we argue for the utility of estimating the distributional drift as the data is being received and show preliminary results of how this could be exploited by a simple replay based method to move beyond the multitask solution.

## 1 INTRODUCTION

Continual learning (CL) (e.g. Ring, 1994; Thrun & Mitchell, 1995; Silver et al., 2013; Parisi et al., 2019; Hadsell et al., 2020; Lesort et al., 2020), sometimes referred to as lifelong learning, directly aims to address the problem of how to construct a model that continuously adapts. Typically, the problem definition — or rather the solution definition — comes down to a list of desiderata that is expected from the system (e.g Schwarz et al., 2018; Hadsell et al., 2020; Mundt et al., 2023). The debate on the ultimate goal of continual learning and the problem definition is still ongoing. In this work we take the view of Mundt et al. (2023): the model needs to be able to remember previous knowledge, hence to deal with *catastrophic forgetting* (McCloskey & Cohen, 1989; French, 1999), and reuse this knowledge to learn quickly new tasks (*forward transfer*), under the assumption that the model capacity is finite and fixed, and the amount of compute it can do per time step is finite and fixed. Traditionally, fixing *catastrophic forgetting* has been seen as the first step towards solving continual learning, as retaining *some* information is needed in order to exhibit transfer, and most research focused on resolving this specific aspect. In this work we question this goal, formally asking whether minimizing catastrophic forgetting is a good objective to achieve continuous adaptation. Our question is inspired by Kumar et al. (2023), who show theoretically that an agent with limited capacity must dynamically compromise between retaining old information and acquiring new information in order to maximise its *lifelong performance* (formalised in Section 3). In other words, minimizing forgetting alone might not achieve the other desiderata of continual learning (e.g. Wołczyk et al., 2021; Wu et al., 2023; Mundt et al., 2023). To understand this trade-off we start by arguing that most methods aimed at solving catastrophic forgetting rely, implicitly or explicitly, on the assumption that a *multi-task* objective is optimal and effectively employ objectives which approximate the multi-task objective. However, depending on the non-stationarity of the data, there can be interference during learning that can make a multi-task objective considerably sub-optimal (e.g. He et al., 2019; Du et al., 2018). Figure 1 depicts this intuition.

Drawing inspiration from the online learning literature, in this work we quantify optimality using the *average lifelong error*, which aligns closely with the concept of dynamic regret, as further elaborated below. In order to study the optimality of the multi-task objective we design two agents: *single-task* (ST) and *multi-task* (MT). The ST agent forgets everything after each task, while the MT agent minimizes the multi-task objective, i.e., the average loss over all previous tasks, and represents a

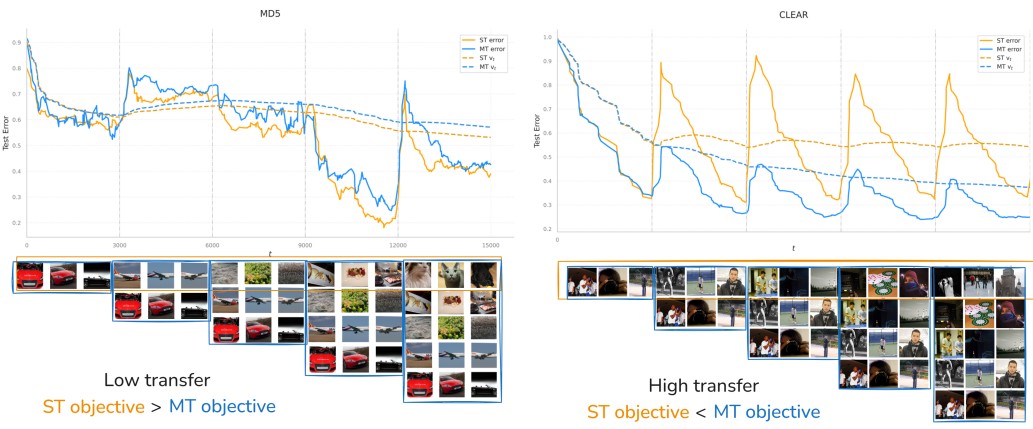

Figure 1: Diagram depicting the potential sub-optimality of the multi-task objective depending on the data distribution. On the right, a data stream is selected such that multitask objective (blue) outperforms single task learner (orange), while on the left the reverse is true. In section 4 we will formalize this behaviour, and in section 5 we will argue that CL algorithms can estimate in which condition they might be and adapt to it.

CL agent with minimal catastrophic forgetting. We present a theoretical and empirical study of the difference between ST and MT agents. Our key contribution is to prove that there exist scenarios where the MT agent accumulates higher regret than the naive forgetting (ST) agent. Furthermore, we demonstrate the extent of this phenomenon across a range of popular supervised learning and reinforcement learning benchmarks. In other words, we effectively prove that *minimizing forgetting does not always result in higher lifelong performance* and that, in some cases, *forgetting can be beneficial for adapting to a changing environment*. These findings validate the thesis of Kumar et al. (2023) in realistic settings, underscoring the nuanced trade-offs intrinsic in continual learning.

The main message of this paper is that the effectiveness of multitask learning is not universal but highly dependent on the nature of the data stream and on the distributional drift during training. This underscores *the importance of considering the specific properties of the data stream* when selecting learning strategies in CL, or even to try to estimate these properties and adapt the CL algorithm as data becomes available. Our results indicate that, when the goal is to maximize the lifelong performance of the agent, the optimal type of agent is inherently data dependent.

## 2 BACKGROUND: FROM MULTI-TASK TO ONLINE LEARNING

Multitask learning (Caruana, 1997) refers to a learning process that averages the losses incurred on multiple tasks. The original goal was to promote sharing of features and therefore speed up learning and resulting in solutions that generalize better. Within the Continual Learning literature the multitask objective comes about when analyzing the ability of systems to prevent *catastrophic forgetting* (McCloskey & Cohen, 1989; French, 1999) and it is consequently incorporated into several existing algorithms, either explicitly or implicitly.

Traditionally — see e.g. Parisi et al. (2019) — continual learning methods tend to be grouped into three categories, according to how they approach the catastrophic forgetting problem, though this categorization is not without fault (see e.g. Titsias et al., 2019). The first category encompasses regularization based methods, such as Elastic Weight Consolidation (EWC) (Kirkpatrick et al., 2017). EWC explicitly assumes the multitask solution as optimal[1], and builds the method as an approximation of this objective when one does not have access to other tasks. This multitask approximation is prevalent, even if sometimes implicitly, in many other regularization methods (e.g. Zenke et al., 2017a; Maltoni & Lomonaco, 2018; Swaroop et al., 2019; Li & Hoiem, 2017, etc.) as recently argued by Yin et al. (2020) and Lanzillotta et al. (2024). The second category of methods consist of replay

---

[1]See equation (2) of their derivation.

methods (e.g. Robins, 1995; Shin et al., 2017), where the replay is effectively emulating the multi-task objective by representing the task not currently available.[2] The third category, dynamic architecture methods (e.g. Zhou et al., 2012; Rusu et al., 2016; Mallya & Lazebnik, 2018) avoid catastrophic forgetting by increasing the capacity of the model. While these methods do not seem to directly mimic the multitask objective, they effectively partition the model capacity between the tasks, which is akin to maximizing the average performance under a fixed capacity constraint. In Appendix B we review some of the most famous algorithms in greater detail, providing evidence for our claims. In general, most continual learning algorithms do not employ a multitask objective; however they can be interpreted as biased estimates thereof. In this work we choose to look at the multitask objective as an abstraction of any specific continual learning algorithm, in order to provide a high level intuition and formalism which can be useful more broadly for the CL community.

Online Learning (OL) on the other hand, (Cesa-Bianchi & Lugosi, 2006; Hoi et al., 2018; Orabona, 2019) offers a fundamentally different perspective on lifelong learning. OL prioritizes rapid adaptability to new data over maintaining strong performance on previously seen data. In this paradigm, algorithms are commonly evaluated using *regret*, a measure that captures the model's ability to adapt efficiently to the evolving data stream throughout its lifetime. This emphasis on adaptability highlights OL's unique approach to addressing the challenges of dynamic environments. In this work we study a common metric in OL known as the *Dynamic Regret* (Herbster & Warmuth, 1998; Zinkevich, 2003) which compares, at each step of the learning, the current expected cost (or reward) of the agent with the minimal achievable cost (or maximal reward). This metric is particularly relevant in slowly-drifting or piecewise stationary settings such as those typically arising in CL (e.g. Hadsell et al., 2020). More precisely, we ignore the comparator and study instead the *average lifelong error* without loss of generality[3].

In continual learning, the adoption of OL metrics is not a new concept. In the context of *Online Continual Learning* (OCL) (Cai et al., 2021; Lopez-Paz & Ranzato, 2017; Aljundi et al., 2019; Buzzega et al., 2020), continual learning algorithms are often evaluated using an online metric. For instance, the *average online accuracy metric* $\mathsf{a}_o$ (Cai et al., 2021) is directly related to the average lifelong error $\mathsf{v}$, with $\mathsf{a}_o = 100 \times (1 - \mathsf{v})$. However, the OCL setting typically assumes both training and evaluation occur in an online manner. this differs from the perspective we adopt in this work. We decouple the training and evaluation protocols, allowing for potentially offline objectives and optimization procedures (i.e., revisiting the same data multiple times), while maintaining an online evaluation of the model's performance. This approach enables us to explore a fundamental question: is minimizing forgetting the right objective for achieving lifelong adaptability?

## 3 SETUP: THE AVERAGE LIFELONG ERROR

In the typical continual learning setting, the agent has to solve a *sequence of tasks*. We consider learning tasks including a target, which broadly covers supervised learning (targets are labels) and reinforcement learning (targets are actions and rewards). For each learning task $\kappa \in \{1, .., K\}$, the agent receives a dataset $D_\kappa = \{(x_1, y_1), \cdots, (x_{N_\kappa}, y_{N_\kappa})\} \sim \mathcal{D}_\kappa$ and learns to predict $Y|X$ through the parametric map $f_{\boldsymbol{\theta}} : \mathcal{X} \to \mathcal{Y}$. For a task $\kappa$ the train error is $R_\kappa(\boldsymbol{\theta}) = {}^1\!/\!_{N_\kappa} \sum_{(x,y) \in D_\kappa} \ell(\boldsymbol{\theta}; \boldsymbol{x}, y)$ and the *test error*, $\mathcal{R}_\kappa = \mathbb{E}_{(x,y) \sim \mathcal{D}_\kappa} [\ell(\boldsymbol{\theta}; x, y)]$. We consider iterative agents with $h$ update steps in each task, such that its *lifetime*[4] is $T = hK$ and we track the (discrete) parameters dynamics $\theta(t)$ along the trajectory.

Our work proposes to compare two types of agents, a *Single Task* (ST) and a *Multi Task* (MT) agent with associated parameter dynamics $\theta_{ST}(t), \theta_{MT}(t)$. An ST agent optimizes the present task loss $R_\kappa$, and is *reset* after completing each task, effectively *forgetting everything*. It serves as a baseline for evaluating performance without employing any continual learning strategies. In contrast, the MT (Multi-Task) agent optimizes the average error across all tasks encountered up to the current point [5], ${}^1\!/\!_\kappa (R_1 + \cdots + R_\kappa)$, without considering future tasks $[\kappa + 1, K]$. Notably, our MT agent differs from traditional multi-task approaches, as it does not have access to information about future tasks.

---

[2]Replay emulates a weighted average objective, where the weight of each task may change with time.

[3]Our derivations can equivalently be applied to dynamic regret.

[4]Our analysis can be extended without difficulty to tasks of various lengths $h_1, ..., h_K$.

[5]Our MT agent does not have access to future tasks as opposed to traditional MT approaches.

Concretely, our goal is to compare the performance of these two types of agents by evaluating the differences in their respective *average lifelong error*:

$$\mathsf{v} = \frac{1}{T} \sum_{i=1}^{K} \sum_{t=(i-1)h+1}^{ih} \mathcal{R}_i\left(\theta(t)\right) \tag{1}$$

To do so, we define $\Delta_T = \mathsf{v}_T^{ST} - \mathsf{v}_T^{MT}$, as the difference in average lifelong error of the two agents. This quantity is central to our study.

Informally, $\Delta_T$ measures the difference in the rate at which the risk on the current task decreases during training. An agent that achieves low risk early in training will have a lower average lifelong error compared to one that achieves a better final performance but at a slower pace. In this context, the ST agent benefits when there is significant "variation" in the task sequence, as the average MT objective may inadequately prioritize the current task. Conversely, when the number of updates per task is severely limited, the MT agent's bias toward averaging across tasks can lead to a lower overall error, provided the tasks are reasonably similar. In other words, $\Delta_T$ captures the trade-off between *stability* and *plasticity* — or bias and variance — in a data-dependent fashion.

**Gradient Descent agents.** In our theoretical analysis, we consider ST and MT agents that update their parameters sequentially using gradient descent (GD) on their respective objectives, with a fixed learning rate $\eta$. In line with the setting described above, the ST agent is reset to some $\boldsymbol{\theta}_0$ at the first step of each task, while the MT agent is not reset, although its objective is updated. Crucially, we do not assume that gradient descent is run to convergence. Instead, the number of update steps per task, $h$, plays a pivotal role in our analysis. As we will demonstrate, $h$ can determine which agent performs best.

## 4 MULTITASK IS NOT ALWAYS OPTIMAL

The primary result of this section demonstrates that, for sufficiently long tasks, the ST agent can outperform the MT agent on non-stationary task sequences where interference between tasks occurs. We formalize this finding in the specific context of convex losses for a linear regression task.

### 4.1 INSTABILITY AND CRITICAL TASK DURATION

Let $\boldsymbol{\theta}_i^\star$ and $\boldsymbol{\theta}_{[1,i]}^\star$ represent the minimizers of the respective ST and MT objectives during task $i$, and define $t_0^i := h(i-1)$ (see Appendix A.1.2 for an exact formula of $\boldsymbol{\theta}_i^\star$ and $\boldsymbol{\theta}_{[1,i]}^\star$ in linear regression.) Notably, our metric of interest can be expressed as:

$$\Delta_T = \frac{1}{K} \sum_{i=1}^{K} \frac{1}{h} \sum_{t=t_0^i+1}^{ih} \underbrace{\left(\mathcal{R}_i(\theta_{ST}(t)) - \mathcal{R}_i(\boldsymbol{\theta}_i^\star)\right)}_{\Delta_T^{ST}} - \underbrace{\left(\mathcal{R}_i(\theta_{MT}(t)) - \mathcal{R}_i(\boldsymbol{\theta}_{[1:i]}^\star)\right)}_{\Delta_T^{MT}} - \underbrace{\left(\mathcal{R}_i(\boldsymbol{\theta}_{[1,i]}^\star) - \mathcal{R}_i(\boldsymbol{\theta}_i^\star)\right)}_{\Delta_T^{I}}$$

Here, we conveniently added and subtracted the risk at the optimal values that these respective agents seek. This introduces an agent-independent term, $\Delta_T^I$, which is unaffected by the choice of agents and instead quantifies the non-stationarity of the learning problem. We refer to this term as *instability*.

We aim to identify the key factors influencing the forgetting vs. no-forgetting trade-off by establishing conditions under which $\Delta_T < 0$, i.e., $\Delta_T^{ST} < \Delta_T^{MT} + \Delta_T^I$. Note that $\Delta_T < 0$ indicates that the single-task agent has a lower *average lifelong error* (i.e., performs better) than the multitask agent.

A critical observation is that the multitask agent benefits from a long sequence of tasks, as evidenced by the fact that $\|\boldsymbol{\theta}_{[1,\kappa-1]}^\star - \boldsymbol{\theta}_{[1:\kappa]}^\star\|_{\boldsymbol{\Sigma}_x^\kappa}^2$ decreases with increasing $\kappa$, so in general, in convex settings [6], $\frac{1}{K}\Delta_T^{MT} \in o(1)$ (see Lemma 9 for a formal proof). Thus, for $K \gg 1$, it is both sufficient and efficient to focus on scenarios where $\Delta_T^{ST} < \Delta_T^I$. In what follows, we adopt a prescriptive view, emphasizing the task duration $h$, as it is a parameter often within the agent's control.

---

[6]It can be verified in experiments that $\Delta_T^{MT}$ decreases with $K$. Please see Table 2.

Proposition 1 defines the minimum task duration required for the single-task agent to match or outperform the multitask learner. In the convex case, using linear models we can prove that such a task duration exists and is finite (see Theorem 4), as long as the instability of the sequence is strictly positive. While the non-linear case can not be approached theoretically,we will later demonstrate that this concept remains empirically useful in such scenarios.

**Proposition 1** (Critical task duration). *The critical task duration $\bar{h}$ is the minimum task duration such that $\Delta_T^{ST} \leq \Delta_T^I$ for all $h > \bar{h}$, where $T = hK$.*

### 4.2 LINEAR PREDICTION WITH CONVEX LOSS

We model each task as a noiseless linear regression problem, where for each task we have $y = \boldsymbol{\theta}_\kappa^{\star\top} \boldsymbol{x}$. The loss function used is the *squared error*, defined as $\ell_2(\boldsymbol{\theta}; \boldsymbol{x}, y) = (\boldsymbol{\theta}^\top \boldsymbol{x} - y)^2$. Consequently, the train and test errors are expressed as follows:

$$R_\kappa(\boldsymbol{\theta}) = (\boldsymbol{\theta} - \boldsymbol{\theta}_\kappa^\star)^\top \hat{\boldsymbol{\Sigma}}_x^\kappa (\boldsymbol{\theta} - \boldsymbol{\theta}_\kappa^\star) \qquad \mathcal{R}_\kappa(\boldsymbol{\theta}) = (\boldsymbol{\theta} - \boldsymbol{\theta}_\kappa^\star)^\top \boldsymbol{\Sigma}_x^\kappa (\boldsymbol{\theta} - \boldsymbol{\theta}_\kappa^\star) \qquad (2)$$

where $\boldsymbol{\Sigma}_x^\kappa = \mathbb{E}_{\boldsymbol{x} \sim \mathcal{D}_\kappa(X)}[\boldsymbol{x}\,\boldsymbol{x}^\top]$ and $\hat{\boldsymbol{\Sigma}}_x^\kappa = \frac{1}{N_\kappa} \sum \boldsymbol{x}_i\,\boldsymbol{x}_i^\top$ are respectively the true and empirical (uncentered) covariance matrices.

**Assumption 2** (Strictly convex losses). For any $\kappa \in [1, K]$ and $M > m > 0$ the spectrum of the covariance matrix satisfies the following condition: $m\,\boldsymbol{I} \preccurlyeq \hat{\boldsymbol{\Sigma}}_x^\kappa \preccurlyeq M\,\boldsymbol{I}$.

Under Assumption 2, GD is known to converge exponentially fast (Boyd & Vandenberghe, 2004). See Lemma 5 for a formal statement. In this case, the ST learner admits the following closed-form expression: within task $i$, the parameter update is given by $\theta_{ST}(t) = \boldsymbol{\theta}_i^\star + (\boldsymbol{I} - \eta\hat{\boldsymbol{\Sigma}}_x^i)^{t-t_0^i} (\boldsymbol{\theta}_0 - \boldsymbol{\theta}_i^\star)$. Since the number of steps per task $h$ is limited, we can *tightly bound* the total error of the ST agent using Assumption 2 and the closed form formula of geometric series:

$$\Delta_T^{ST} = \frac{1}{K} \sum_{i=1}^K \frac{1}{h} \sum_{t=1}^h \|\boldsymbol{\theta}_0 - \boldsymbol{\theta}_i^*\|_{\boldsymbol{\Sigma}_x^\kappa}^2 (1 - \eta\hat{\boldsymbol{\Sigma}}_x^i)^{2t} \in \Theta\left( \frac{1}{K} \sum_{i=1}^K \frac{\|\boldsymbol{\theta}_0 - \boldsymbol{\theta}_i^*\|_{\boldsymbol{\Sigma}_x^\kappa}^2}{h} \frac{1 - \epsilon^h}{1 - \epsilon} \right)$$

where $\epsilon = (1 - \eta m)^2$ in the upper bound and $\epsilon = (1 - \eta M)^2$ in the lower bound. This expression leads to a tight bound on the lifelong error difference $\Delta_T$, as presented in Theorem 13 in the appendix. Consequently, we establish a crucial first result of our study.

**Corollary 3** (Monotonic dependence on task duration). *For a suitable choice of learning rate and a fixed task duration $h$, gradient descent on the ST and MT convex objectives described in Section 4.2 gives rise to two parameter dynamics, $\theta_{ST}(t)$ and $\theta_{MT}(t)$, such that $\Delta_T$ decreases monotonically with the task duration.*

The task duration $h$ is typically controlled by the agent designer. As a consequence of Corollary 3, increasing the task duration will necessarily decrease the difference $\Delta_T$. However (Corollary 12 in the Appendix) it is not granted that increasing $h$ will ever result in $\Delta_T < 0$, i.e. that a critical task duration exists in general. Our main result guarantees the existence of a critical task duration, when the instability of the sequence is *strictly positive*. An informal version of the theorem is stated here, with the formal treatment detailed in the Appendix.

**Theorem 4** (Existence result, informal). *In the same setting as Corollary 3, if the instability of the sequence is positive then there exists a finite critical task duration $\bar{h}$.*

This result arises from solving for $h$ in the bound for $\Delta_T$, yielding a threshold value $\hat{h} < \infty$, with $\bar{h} \leq \hat{h}$ by definition. In other words, Theorem 4 proves that the MT objective is not *always* optimal with respect to the average lifelong error. Instead, long tasks or highly non-stationary problems may be better solved by an ST agent. Conversely, our study also proves that there are cases where the ST agent is not optimal either, specifically when $\Delta_T > 0$. As a consequence, *the choice of agent should depend on the specific problem*, if the goal is to minimize the average lifelong error.

While we have the full extent of our study provided in the Appendix, we summarize the key findings here: (1) that $\Delta_T$ decreases monotonically as the task duration $h$ increases (Corollary 3); (2) if the

instability $\Delta_T^I > 0$, then there exists a finite critical task duration (Theorem 4); (3) increasing $\Delta_T^I$ decreases the critical task duration (Theorem 16).

In the remainder of the paper, we assess to what extent these findings extend to the more complex setting of neural network training, evaluating the behaviour of ST and MT agents on popular supervised learning and reinforcement learning benchmarks.

**A note on overparametrization.** Assumption 2 implies that the system is not overparametrized, i.e. $p < N_\kappa$ for all $\kappa$. In order to deal with the overparametrized case it is sufficient to add a norm regularizer $\lambda \|\boldsymbol{\theta}\|^2$ to the loss in our derivations. This minor modification can be seamlessly integrated into our derivations without affecting the results, as we show in Appendix A.4.

### 4.3 ILLUSTRATION ON A SIMPLE SETTING

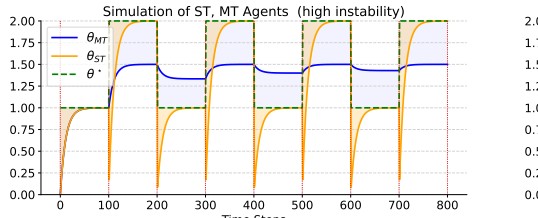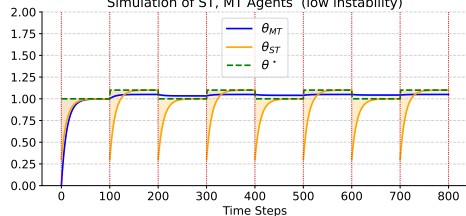

Figure 2: Toy Settings comparisons. $\theta^\star$ oscillates between $1$ and $2$ for each task on the left, and between $1$ and $1.1$ for each task on the right. There are $8$ tasks (with start marked by dashed red lines) with $h = 100$ each, and $\eta = 0.01$. Both agents are initialized with $\theta_0 = 0$. The shaded area corresponds to the lifelong error of the agent.

In order to build a concrete intuition for the theoretical results we look into two toy settings, depicted in Figure 2. The tasks in the figure are one-dimensional and two different tasks with optimal solutions $\theta_1^\star$ and $\theta_2^\star$ (in green) occur repeatedly in alternating fashion. In the first case (on the left) the difference between the two solutions is $1$ and in the second case (on the right) the difference is only $0.1$. For the convex least-square setting, i.e $\forall i, \mathcal{R}_i(\theta) = \sigma^2 (\theta - v_i)^2$, the instability is a function of the difference between the two solutions (full derivations in Appendix A.3):

$$\frac{1}{K} \Delta_T^I = \frac{\sigma^2}{2} (\theta_1^\star - \theta_2^\star)^2 \tag{3}$$

As expected, the instability is higher when the difference between task solutions is more pronounced, as seen in the left-hand figure. According to Theorem 4, a critical task duration exists for both tasks, given that instability remains strictly positive in both scenarios. From Corollary 3, it follows that, with all other factors held constant, the critical task duration is expected to be lower in the left-hand toy setting. This is evident as, despite using the same task duration of $h = 100$ in both cases, the ST agent accumulates less error over time in the first scenario, whereas the MT agent demonstrates better average performance in the second. In Appendix A.3, we simulate the evolution of $\Delta_T$ by varying the duration $h$, and confirm empirically that the critical task duration is approximately half in the first toy setting.

## 5 EMPIRICAL ANALYSIS

Our empirical analysis is structured into three main parts. First, we validate our theoretical framework on complex continual learning benchmarks, encompassing both supervised learning and reinforcement learning tasks. Next, we turn to a toy benchmark, Permuted-CIFAR10, where we can control the task sequence's instability by adjusting the permutation strength. This setup enables us to test our theoretical predictions regarding the relationship between instability and task duration. Finally, we showcase the practical applicability of our framework in continual learning by implementing a simple variant of experience replay, where the objective is tailored to the instability of the data stream.

### 5.1 SINGLE TASK VS MULTI TASK IN THE WILD

**Benchmarks** We present results for supervised learning and reinforcement learning benchmarks. For supervised learning we take two different benchmarks: the first is based on the CLEAR dataset (Lin et al., 2021), a collection of images of 10 different classes spanning the years 2004-2014. We split the collection into 10 tasks, one for each year. The second benchmark is a sequence of 5 different open source classification datasets, with no semantic overlap between them. In particular, the tasks consists in classification of automobile models (Krause et al., 2013), aircraft models (Maji et al., 2013), textures (Cimpoi et al., 2014), dishes (Bossard et al., 2014) and pets (Parkhi et al.). Each dataset has originally a different number of classes, samples and a different input size. To avoid introducing biases in the models, we standardize all tasks to have only 30 classes, and we use the same batch size and amount of update steps in each task, regardless of the original dataset size. Hereafter we refer to this as the "MULTIDATASET" (MD5) benchmark. We have chosen these two benchmarks because they represent different types of distribution shifts. While the transitions from one task to the next in CLEAR are arguably smooth (the tasks differ in input resolution but semantically are equivalent), in MD5 they are sharp, changing the semantics of the task altogether. For reinforcement learning we rely on the Meta-World (MW) benchmark (Yu et al., 2020), which is a collection of 50 distinct simulated robotic manipulation environments. We train our agents on a sub-collection of 10 environments called ML10 and we evaluate their average lifelong reward on the same environments in an online fashion. We chose this environment due to it being previously used to highlight interference in continual learning (Wołczyk et al., 2021). More details in Appendix D.

**Notes on the empirical setup.** In line with out theoretical analysis, we use the same task duration $h$ for each task -which is also in line with typical practice in continual learning. More precisely, $h$ is the number of parameter updates performed, which may correspond to multiple passes through the dataset. After each update the performance of the agent is evaluated on a separate test set, in the case of supervised learning, or on new interactions with the environment. The evaluation is always performed on the current task. Additionally, at the end of training on all the tasks we measure the agent's *multitask (offline) accuracy $ACC_{agent}$ or multitask (offline) reward $R_{agent}$*, which consists in the average performance across all tasks, and is a typical CL metric (Lopez-Paz & Ranzato, 2017; Powers et al., 2022). To aid interpretability and comparison with the offline performance we report the average lifelong accuracy $a_o = (1 - v) \times 100$ which is more common in the literature (Cai et al., 2021). More details regarding our experimental choices in Appendix D.

| | $h$ | $a_{o\,ST}$ | $a_{o\,MT}$ | $\Delta_T$ | $ACC_{ST}$ | $ACC_{MT}$ |
|---|---|---|---|---|---|---|
| MD5 | 3000 | $47.0_{\pm 0.002}$ | $43.0_{\pm 0.005}$ | $-0.004_{\pm 0.005}$ | $19.9_{\pm 0.007}$ | $62.8_{\pm 0.007}$ |
| CLEAR | 3000 | $46.5_{\pm 0.0004}$ | $68.1_{\pm 0.0005}$ | $+0.216_{\pm 0.002}$ | $65.2_{\pm 0.012}$ | $76.8_{\pm 0.004}$ |

| | | $r_{o\,ST}$ | $r_{o\,MT}$ | $\Delta_T$ | $R_{ST}$ | $R_{MT}$ |
|---|---|---|---|---|---|---|
| ML10 | 500 | $1.15_{\pm 0.21}$ | $0.77_{\pm 0.30}$ | $-0.38_{\pm 0.19}$ | $1.007_{\pm 0.09}$ | $1.029_{\pm 0.14}$ |

Table 1: Lifelong average accuracy ($a_o$) / reward ($r_o$) and multitask accuracy ($ACC$) / reward ($R$) in the wild. Higher is better. We report the difference in performance $\Delta_T$ in the original metric, e.g $\Delta_T = v_{ST} - v_{MT}$ and $\Delta_T = -(r_{ST} - r_{MT})$. The lower $ACC$ ($R$), the higher the forgetting in supervised (RL) benchmarks.

Table 1 shows the performance of the ST and MT agent on the three benchmarks. *The ST agent outperforms the MT agent according to the lifelong average performance metrics ($a_o/r_o$) in the MD5 and ML10 benchmarks, while the opposite is true in the CLEAR benchmark.* This confirms our intuition that the interference between the tasks is lower in CLEAR, making the multitask a suitable objective. In Section 5.2 we quantify this statement by measuring the amount of instability $\Delta_T^I$ in all our benchmarks. Notice that *the MT agent always outperforms the ST agent on the multitask performance metrics ($ACC/R$), indicating that – as expected – its forgetting is always lower.*

Next, we ask whether increasing the task duration $h$ would reduce the advantage of the multitask agent in CLEAR and ML10, as predicted by the theory. Table 2 shows the behaviour of our performance metrics as the task duration $h$ is increased. *In accordance with the theory, on CLEAR we observe the error difference $\Delta_T$ decaying with $h$*, although it does not fall below 0 -suggesting that the critical task duration may be way above the range of $h$ tested. Interestingly, we also observe that multitask performance of both ST and MT improve as $h$ is increased. The reason is that the similarity of the tasks grants positive transfer between them, and thus improving performance on one task by training

| | $h$ | $\mathsf{a}_{o\,ST}$ | $\mathsf{a}_{o\,MT}$ | $\Delta_T$ | $ACC_{ST}$ | $ACC_{MT}$ |
|---|---|---|---|---|---|---|
| CLEAR | 3000 | 46.5 $_{\pm 0.0004}$ | 68.1 $_{\pm 0.0005}$ | 0.216 $_{\pm 0.002}$ | 65.2 $_{\pm 0.012}$ | 76.8 $_{\pm 0.004}$ |
| | 6000 | 56.2 $_{\pm 0.0001}$ | 71.3 $_{\pm 0.004}$ | 0.151 $_{\pm 0.004}$ | 75.9 $_{\pm 0.017}$ | 78.4 $_{\pm 0.003}$ |
| | 9000 | 61.0 $_{\pm 0.0004}$ | 72.0 $_{\pm 0.001}$ | 0.11 $_{\pm 0.001}$ | 78.1 $_{\pm 0.009}$ | 78.9 $_{\pm 0.010}$ |
| | 12000 | 64.1 $_{\pm 0.0002}$ | 73.2 $_{\pm 0.0006}$ | **0.10** $_{\pm 0.001}$ | 76.9 $_{\pm 0.0008}$ | 79.3 $_{\pm 0.001}$ |

| | | $\mathsf{r}_{o\,ST}$ | $r_{o\,MT}$ | $\Delta_T$ | $R_{ST}$ | $R_{MT}$ |
|---|---|---|---|---|---|---|
| ML10 | 50 | 1.07 $_{\pm 0.10}$ | 0.62 $_{\pm 0.08}$ | -0.45 $_{\pm 0.06}$ | 1.593 $_{\pm 0.12}$ | 0.355 $_{\pm 0.08}$ |
| | 500 | 1.15 $_{\pm 0.21}$ | 0.77 $_{\pm 0.30}$ | -0.38 $_{\pm 0.19}$ | 1.007 $_{\pm 0.09}$ | 1.029 $_{\pm 0.14}$ |

Table 2: Increasing the task duration $h$ in CLEAR and ML10, closes the gap in average lifelong performance.

for longer has the additional effect of increasing the performance on all the other tasks. On the other hand on the ML10 benchmark -where the ST agent consistently outperforms the MT agent on the current task- the reward difference does not decay with $h$. We hypothesise that this might be a result of the inherent noisiness of the reward signal, which we use as a performance metric.

| $K$ | $\mathsf{r}_{o\,ST}$ | $\mathsf{r}_{o\,MT}$ | $\Delta_T$ | $R_{ST}$ | $R_{MT}$ |
|---|---|---|---|---|---|
| 3 | 0.90 $_{\pm 0.37}$ | 0.70 $_{\pm 0.34}$ | -0.21 $_{\pm 0.24}$ | 0.58 $_{\pm 0.06}$ | 0.81 $_{\pm 0.38}$ |
| 6 | 1.02 $_{\pm 0.30}$ | 0.92 $_{\pm 0.48}$ | -0.10 $_{\pm 0.23}$ | 0.48 $_{\pm 0.10}$ | 1.03 $_{\pm 0.78}$ |
| 10 | 1.15 $_{\pm 0.21}$ | 0.77 $_{\pm 0.30}$ | -0.38 $_{\pm 0.19}$ | 1.007 $_{\pm 0.09}$ | 1.029 $_{\pm 0.14}$ |

Table 3: Increasing the number of tasks in ML10. The sequence order is fixed, and the number of tasks $K$ observed is chosen between $3, 6, 10$ (10 corresponds to the full sequence).

Finally, we evaluate the effect of increasing $K$, the number of tasks in the sequence, on the trade-off between forgetting and memorizing. We perform this experiment on the ML10 benchmark, where the tasks are known to be adversarial in nature. We train the ST and MT agents on a sequence of 3, 6 or 10 tasks presented with the same ordering. In Table 3 we present the results. If there are more difficult tasks later in the sequence increasing the number of tasks should lead to increased instability in ML10 experiments. In Table 11 we report the average reward on each task: we observe a marked difference in difficulty between the tasks, with easier tasks appearing later in the sequence. The observed increase in average lifelong rewards in Table 3 reflects the distribution of the difficulty in the task ordering. Tasks that yield higher rewards on average, boost the overall performance. Even though there is no clear monotonic trend of $\Delta_T$, we observe that ST globally outperforms MT on average lifelong reward, which is in line with the fact that the first $K = 3$ tasks have relatively high interference and difficulty.

## 5.2 EMPIRICAL STUDY ON THE CRITICAL TASK DURATION AND INSTABILITY

We move on to study empirically the critical task duration and instability in non-convex settings. According to the theory, the critical task duration depends on the sequence instability $\Delta_T^I$, which is by definition a property of the data, independent of the agents: $\Delta_T^I = {}^1/K \sum_{\kappa=1}^{K} \mathcal{R}_\kappa(\boldsymbol{\theta}_{[1,\kappa]}^\star) - \mathcal{R}_\kappa(\boldsymbol{\theta}_\kappa^\star)$. In convex settings this quantity can be directly measured (see Equation (20) for a precise formula). However when using non-linear models such as neural networks, the task minimizer $\boldsymbol{\theta}_i^\star$ is not known nor easy to discover. Additionally, when using neural networks the notion of task similarity is inherently model dependent since the features representing the data are.

| Data | Option 1 | Option 2 |
|---|---|---|
| CLEAR | $-0.024_{\pm 0.003}$ | $0.007_{\pm 0.001}$ |
| MD5 | $0.017_{\pm 0.008}$ | $0.35_{\pm 0.01}$ |
| ML10 | $0.407_{\pm 0.002}$ | $0.139_{\pm 0.009}$ |
| PC-16 | $-0.0213_{\pm 0.0024}$ | $0.03_{\pm 0.002}$ |
| PC-32 | $0.0014_{\pm 0.004}$ | $0.30_{\pm 0.005}$ |

Table 4: Measures of instability. The higher the measure the higher the instability. The range of values is not the same for supervised and RL benchmarks. We highlight in gray the toy benchmarks.

Hence the question: *how can the instability be estimated in non-convex settings?*

**Option 1** We propose to approximate $\Delta_T^I$ by training a neural network on the ST and MT objectives, obtaining respectively $\tilde{\theta}_i^\star$ and $\tilde{\theta}_{[1,i]}^\star$ and measure $\tilde{\Delta}_T^I = \frac{1}{K} \sum_{i=1}^\kappa (\mathcal{R}_i(\tilde{\theta}_{[1,i]}^\star) - \mathcal{R}_i(\tilde{\theta}_i^\star))$. Note that this quantity is *dependent* on initialization, optimizer and hyperparameters of the experimental setup.

**Option 2** Intuitively, $\Delta_T^I$ should be higher when there is more interference between the tasks and lower when the tasks have more in common. Thus, we propose to measure directly the transfer between tasks as a proxy for instability. More specifically, we produce a *transfer matrix* $\mathcal{Q}$ whose $i, j$ entry is $\mathcal{R}_j(\tilde{\theta}_i^\star)$ and we compare the average of the diagonal to that of the off-diagonal. In practice, this second option is cheaper to compute, as it does not require to train two separate models and it can be estimated online (provided the agent has access to the full sequence).

In Table 4 we report the measurements of instability with both options. In the supervised learning benchmarks we take $\mathcal{R}(\theta)$ to be the test error (thus a quantity between 0 and 1) and in ML10 we use $\mathcal{R}(\theta) = -r(\theta)$, which is generally unbounded. Overall, we observe that the first option can be negative (because $\mathcal{R}_i(\tilde{\theta}_i^\star) \neq 0$ for our choice of $\mathcal{R}$) and the second option is always positive (because training on a task necessarily results in a higher performance on the task, thus $\mathcal{Q}_{ii} < \mathcal{Q}_{ij} \ \forall j \neq i$). Both metrics confirm the intuition that the instability is lower in the CLEAR dataset, and higher in the Md5 and ML10 datasets, which aligns with the observed $\Delta_T$.

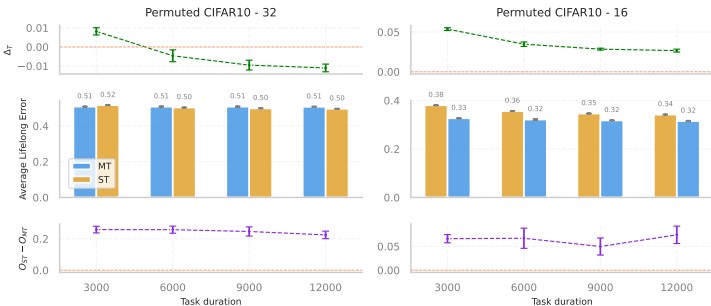

Figure 3: Permuted CIFAR experiments. Top: evolution of $\Delta_T$ as a function of $h$. Middle: average lifelong errors of MT and ST agents as a function of $h$. Bottom: evolution of the multitask performance as a function $h$.

Next, we wish to explore empirically how $\Delta_T^I$ impacts $\bar{h}$, by controlling $\Delta_T^I$ in a toy experimental setting. More specifically, we build a benchmark from the CIFAR 10 data Krizhevsky & Hinton (2009), applying fixed random permutations to the images in the dataset. By increasing the size of the permuted areas of the input image we wish to increase the instability $\Delta_T^I$. We use two different permutation sizes in all experiments, namely 16 and 32. We refer to the respective benchmarks as 'CIFAR10 Permuted - 16' (PC-16) and 'CIFAR10 Permuted - 32' (PC-32).

The instability measures introduced above (Table 4) validate our methodology: for both measures instability is higher for PC-16 than PC-32. In Figure 3 we visualize the average lifelong error v of the ST and MT agents as we increase the task duration $h$. As a comparison, we also visualize the evolution of the difference in multitask performance, which should be independent of $h$. The critical task performance corresponds to the value of $h$ where $\Delta_T$ is predicted to drop below 0. Since $\Delta_T$ is always positive in PC-16, we infer that the critical task duration lays beyond the explored range. However, the critical task duration for PC-32 is estimated to be between 3000 and 6000 steps: as predicted by the theory, higher $\Delta_T^I$ corresponds to lower $\bar{h}$.

### 5.3 DEMO: A DATA-DEPENDENT OBJECTIVE FOR REPLAY

One of the main takeaway messages of this work is that the optimization objective in continual learning should be treated as a data-dependent quantity. Broadly speaking, the objective should reflect the instability of the sequence, enabling forgetting when it is high and avoiding it when it is low.

We design a simple variant of the experience replay (ER) algorithm (Lin, 1992; Zhang & Sutton, 2020), which we call *Selective Replay* (SR) that does not replay from previous tasks when there is high instability in the sequence. Generally, one could rely on any heuristical measure of $\Delta_T^I$ and adapt to the current stream, trading forgetting for forward transfer. In practice, in this simple experiment we

create a new controlled benchmark from CIFAR10, which we call 'C10 mixed', where we increase the permutation size from 16 to 32 after 5 tasks. We know from Figure 3 that forgetting is beneficial when the permutation size is 32, since the instability is very high (we choose $h = 6000$ such that $\Delta_T < 0$). Intuitively, in this benchmark memory is useful only on the first half of the sequence, where there is positive transfer between the tasks. Thus, both the ER and ST agent are suboptimal, as the former is forced to remember irrelevant information -which affects its capacity to fit the new data-and the latter fails to remember any useful information. SR is designed to remember the relevant information and discard irrelevant one. We take advantage of the knowledge of the sequence, and simply change the objective from ER to the ST objective when the instability is increased.

In Figure 4 we plot the test error over the training trajectory of the three agents: ST, ER and SR. As expected, the SR agent has the lowest average lifelong error, and the ER agent has the lowest multitask error - meaning that it has the lowest forgetting. Observe the switch in behaviour midway through the sequence of task: in the first half of the sequence the ER agent outperforms the ST agent, while the opposite is true in the second half. Because of its dynamic objective, the SR agent is able to always adhere to the best performing behaviour.

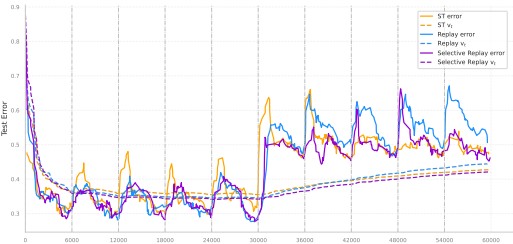

| Agent | $a_o$ | $ACC$ |
|---|---|---|
| ER | 55.1 $_{\pm 0.001}$ | **39.4** $_{\pm 0.01}$ |
| ST | 57.0 $_{\pm 0.001}$ | 24.0 $_{\pm 0.012}$ |
| SR | **58.2** $_{\pm 0.005}$ | 24.0 $_{\pm 0.002}$ |

Figure 4: Cifar 10 mixed results. Left: test error trajectory through training, evaluated on the current task. In violet the SR agent, in blue the classic ER agent and in yellow the ST agent. Right: average lifelong performance and multitask performance at the end of training.

Clearly, crucial to the success of selective replay, and any kind of adaptive objective, is the information regarding the tasks sequence instability -which in the case of this experiment is assumed to be known. Thus, the question becomes how to estimate $\Delta_T^I$ in an online fashion, as the data stream is being processed. We believe that this is an exciting avenue for future research, together with the study of data-dependent objectives.

## 6 DISCUSSION AND CONCLUSION

In this work we explore the *optimality* of the multitask objective in continual learning. Multitask objectives arise as a natural target to address *catastrophic forgetting*. However, as was highlighted in previous works as well, the multitask objective is suboptimal when considering the *overall continual learning problem, which ultimately is about lifelong adaptability.* Borrowing from the rich literature on online learning, we formalize sufficient conditions for suboptimality in the restricted scenario of convex objective and linear models. We show empirically that our theoretical results can be predictive of the behaviour of the nonlinear system. We discuss the limitations of our approach in Appendix C.

Crucially we believe our work highlights at least three different observations. Firstly, while the suboptimality of the multitask objective was observed early on in continual learning, most methods are still heavily relying on it. *We argue that this is not necessary.* Indeed, we showed that one can easily modify a replay based method to take into account task similarity and be able to outperform the multitask agent. We argue that more continual learning methods should remove the reliance on multitask objective or at least reason explicitly about the assumptions being made. Secondly, we argue that without making assumptions on data stream, one cannot behave optimally. Thus, it should be common for continual methods to exploit the structure of the data stream, either estimating online or assuming it as initial condition. Third, in order to do the above, further formalization of the continual learning problem and *theoretical tools to describe data non-stationarity* are needed. In particular, connecting the field with related topics, such as online learning, but also others like invariances, causality, can provide a rich source to borrow from and adapt mathematical constructs.

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

# Appendices

## A  THEORETICAL PROOFS

**Notation**

| | |
|---|---|
| $\mathcal{D}_1, \ldots, \mathcal{D}_K$ | A sequence of tasks: $K$ datasets |
| $\mathcal{X}$ | Input space |
| $\mathcal{Y}_i$ | Task $i \in [K]$ output space (may vary or be shared across tasks) |
| $\boldsymbol{\theta} \subseteq \mathbb{R}^P$ | The neural network parameters |
| $\boldsymbol{I}_n$ | Identity matrix with $n$ rows and $n$ columns |
| $\boldsymbol{\theta}$ | A generic network parameters vector |
| $\boldsymbol{\theta}_{\texttt{Agent}}(t)$ | Dynamics of the network parameters of `Agent` (ST,MT) |
| $\boldsymbol{\theta}_0$ | Network initialization |
| $\ell_i(x, y, \boldsymbol{\theta})$ | Task $i$ loss function |
| $\mathcal{R}_i(\boldsymbol{\theta})$ | Expected loss on the task $i$ distribution $\mathcal{D}_i$ |
| $R_t(\boldsymbol{\theta})$ | Empirical loss on the task $i$ dataset |
| $t_0^\kappa$ | First time step of task $\kappa$, equal to $(\kappa - 1)\, h$ since all tasks last $h$ time steps |
| $\|x\|_{\boldsymbol{\Sigma}} = x^\top \boldsymbol{\Sigma} x$ | Elliptic norm of vector $x$ for PSD matrix $\boldsymbol{\Sigma}$ |

### A.1  RECALL THE SETUP

Average lifelong error:

$$\frac{1}{T} \sum_{t=1}^{T} \mathbb{E}_{x_t, y_t} \ell(\,\theta(t); x_t, y_t\,) = \frac{1}{T} \sum_{i=1}^{K} \sum_{t=1}^{h_i} \mathcal{R}_i(\,\theta(t_0^i + t)\,) \tag{4}$$

Average lifelong error difference:

$$\frac{1}{T} \sum_{i=1}^{K} \sum_{t=1}^{h_i} \left( \mathcal{R}_i(\,\theta_{ST}(t_0^i + t)\,) - \mathcal{R}_i(\,\theta_{MT}(t_0^i + t)\,) \right) \tag{5}$$

Agents' objectives:

$$\Omega_{ST}(\boldsymbol{\theta}, \kappa) = R_\kappa(\boldsymbol{\theta}) \qquad \Omega_{MT}(\boldsymbol{\theta}, \kappa) = \frac{1}{\kappa} \sum_{i=1}^{\kappa} R_i(\boldsymbol{\theta}) \tag{6}$$

#### A.1.1  LINEAR REGRESSION MODEL

We define each task as a linear regression problem:

$$y = \boldsymbol{\theta}_\kappa^{\star\top} \boldsymbol{x} + \xi \tag{7}$$

where $\xi$ is a noise term sampled independently for each input $\boldsymbol{x}$ with mean $0$ and variance $\boldsymbol{\Sigma}^2$. In the paper we treat the noiseless case, i.e. assume $\xi = 0$. For completeness, we keep the setting formulation more general.

Let $\mathcal{D}_\kappa(X)$ the marginal distribution on the input space $\mathcal{X}$ and $D_\kappa$ a dataset of size $N_\kappa$ sampled i.i.d. from $\mathcal{D}_\kappa$. We denote by $\boldsymbol{\Sigma}_x^\kappa = \mathbb{E}_{x \sim \mathcal{D}_\kappa(X)}[\boldsymbol{x}\boldsymbol{x}^\top]$ the uncentred *population or true covariance* matrix of the inputs $x$. Given a training dataset of size $N_\kappa$ for task $\kappa$ we define the *empirical covariance* matrix as $\hat{\boldsymbol{\Sigma}}_x = \frac{1}{N_\kappa} \sum_{i=1}^{N_\kappa} x_i x_i^\top$.

With a squared error $\ell_2(\boldsymbol{\theta}; \boldsymbol{x}, y) = (\boldsymbol{\theta}^\top \boldsymbol{x} - y)^2$ the risk or test error $\mathcal{R}_\kappa(\boldsymbol{\theta})$ of the predictor $f_{\boldsymbol{\theta}} = \boldsymbol{\theta}^\top \boldsymbol{x}$ is:

$$\begin{aligned}
\mathcal{R}_\kappa(\boldsymbol{\theta}) &= \mathbb{E}_{x,\xi}\left[\langle \boldsymbol{\theta}_\kappa^\star - \boldsymbol{\theta}, \boldsymbol{x}\rangle - \xi\right]^2 \\
&= (\boldsymbol{\theta} - \boldsymbol{\theta}_\kappa^\star)^\top \boldsymbol{\Sigma}_x (\boldsymbol{\theta} - \boldsymbol{\theta}_\kappa^\star) + \sigma^2
\end{aligned} \tag{8}$$

Similarly, the training error is simply:

$$\begin{aligned}
R_\kappa(\boldsymbol{\theta}) &= \frac{1}{N_\kappa} \sum_{i=1}^{N_\kappa} \left[(\boldsymbol{\theta}_\kappa^\star - \boldsymbol{\theta})^\top \boldsymbol{x}_i - \xi_i\right]^2 \\
&= (\boldsymbol{\theta}_\kappa^\star - \boldsymbol{\theta})^\top \left(\frac{1}{N_\kappa} \sum_{i=1}^{N_\kappa} \boldsymbol{x}_i \boldsymbol{x}_i^\top\right)(\boldsymbol{\theta}_\kappa^\star - \boldsymbol{\theta}) - \frac{2}{N_\kappa} \sum_{i=1}^{N_\kappa} \xi_i \, \boldsymbol{x}_i^\top (\boldsymbol{\theta}_\kappa^\star - \boldsymbol{\theta}) \\
&= (\boldsymbol{\theta}_\kappa^\star - \boldsymbol{\theta})^\top \hat{\boldsymbol{\Sigma}}_x^\kappa (\boldsymbol{\theta}_\kappa^\star - \boldsymbol{\theta}) - \frac{2}{N_\kappa} \sum_{i=1}^{N_\kappa} \xi_i \, \boldsymbol{x}_i^\top (\boldsymbol{\theta}_\kappa^\star - \boldsymbol{\theta}) \\
&\overset{\xi_i = 0 \forall i}{=} (\boldsymbol{\theta}_\kappa^\star - \boldsymbol{\theta})^\top \hat{\boldsymbol{\Sigma}}_x^\kappa (\boldsymbol{\theta}_\kappa^\star - \boldsymbol{\theta}) \\
&= (\boldsymbol{\theta}_\kappa^\star - \boldsymbol{\theta})^\top \boldsymbol{\Sigma}_x (\boldsymbol{\theta}_\kappa^\star - \boldsymbol{\theta}) - \left((\boldsymbol{\theta}_\kappa^\star - \boldsymbol{\theta})^\top (\boldsymbol{\Sigma}_x - \hat{\boldsymbol{\Sigma}}_x^\kappa)(\boldsymbol{\theta}_\kappa^\star - \boldsymbol{\theta})\right) \\
&= \mathcal{R}_k(\boldsymbol{\theta}) + \left((\boldsymbol{\theta}_\kappa^\star - \boldsymbol{\theta})^\top (\boldsymbol{\Sigma}_x - \hat{\boldsymbol{\Sigma}}_x^\kappa)(\boldsymbol{\theta}_\kappa^\star - \boldsymbol{\theta})\right)
\end{aligned} \tag{9}$$

where in the last line, we highlight that in this simple setting, the training error is equal to the test error up to a vanishing error term that goes to 0 as $N_\kappa$ grows large. This result is standard and typical of empirical risk minimization (Vapnik, 1991). More precisely, the norm of the error decreases in $O(1/\sqrt{N_k})$ (with a hidden constant factor that depends on the spectrum of $\boldsymbol{\Sigma}_x$).

Finally, notice that in the noiseless case $\mathcal{R}_\kappa(\boldsymbol{\theta}_\kappa^\star) = 0$ by Equation (8).

**Assumption 2.** *For any $\kappa \in [1, K]$ and $M > m > 0$ the spectrum of the covariance matrix satisfies the following condition:*

$$m \, \boldsymbol{I} \preccurlyeq \hat{\boldsymbol{\Sigma}}_x^\kappa \preccurlyeq M \, \boldsymbol{I}$$

### A.1.2 Minimizers

Given a sequence of $K$ tasks we can resolve for the minimizers of, respectively, the MT and ST objectives. Trivially, $\arg\min_{\boldsymbol{\theta}} \Omega_{ST}(\boldsymbol{\theta}, \kappa) = \boldsymbol{\theta}_\kappa^\star$. For MT we have:

$$\begin{aligned}
\boldsymbol{\theta}_{[1,\kappa]}^\star &:= \arg\min_{\boldsymbol{\theta}} \Omega_{MT}(\boldsymbol{\theta}, \kappa) \\
&= \arg\min_{\boldsymbol{\theta}} \sum_{i \le \kappa} (\boldsymbol{\theta}_i^\star - \boldsymbol{\theta})^\top \hat{\boldsymbol{\Sigma}}_x^i (\boldsymbol{\theta}_i^\star - \boldsymbol{\theta}) = \left(\sum_{i \le \kappa} \hat{\boldsymbol{\Sigma}}_x^i\right)^{-1} \left(\sum_{i \le \kappa} \hat{\boldsymbol{\Sigma}}_x^i \boldsymbol{\theta}_i^\star\right)
\end{aligned}$$

For simplicity, we denote $\sum_{i \le \kappa} \hat{\boldsymbol{\Sigma}}_x^i$ by $\bar{\boldsymbol{\Sigma}}_{\bar{x}}^{\le \kappa}$ and $\sum_{i \le \kappa} \hat{\boldsymbol{\Sigma}}_x^i \boldsymbol{\theta}_i^\star$ by $\bar{\boldsymbol{\theta}}_{[1,\kappa]}^\star$.

### A.1.3 Gradient descent dynamics

The ST agent and MT agent update their parameters by gradient descent on their respective objectives with a learning rate $\eta$. We here consider the case of full batch gradient descent. One iteration during task $\kappa$ takes the form:

$$\begin{aligned}
\theta_{ST}(t) &\leftarrow \theta_{ST}(t-1) - \eta \nabla_{\theta_{ST}(t-1)} R_\kappa(\boldsymbol{\theta}) \\
&= \theta_{ST}(t-1) - \eta \, \hat{\boldsymbol{\Sigma}}_x^\kappa (\theta_{ST}(t-1) - \boldsymbol{\theta}_\kappa^\star)
\end{aligned} \tag{10}$$

$$\begin{aligned}
\theta_{MT}(t) &\leftarrow \theta_{MT}(t-1) - \frac{\eta}{\kappa} \sum_{i=1}^{\kappa} \nabla_{\theta_{MT}(t-1)} R_i(\boldsymbol{\theta}) \\
&= \theta_{MT}(t-1) - \frac{\eta}{\kappa} \sum_{i \le \kappa} \hat{\boldsymbol{\Sigma}}_x^i (\theta_{MT}(t-1) - \boldsymbol{\theta}_i^\star)
\end{aligned} \tag{11}$$

Let $t_0^\kappa$ be the beginning of task $\kappa$ and $t$ the absolute time step. Solving the recursion we have:

$$\theta_{ST}(t) = \boldsymbol{\theta}_\kappa^\star + (\boldsymbol{I} - \eta\hat{\boldsymbol{\Sigma}}_x^\kappa)^{(t-t_o^\kappa)}(\boldsymbol{\theta}_0 - \boldsymbol{\theta}_\kappa^\star) \tag{12}$$

$$\theta_{MT}(t) = \boldsymbol{\theta}_{[1,\kappa]}^\star + (\boldsymbol{I} - \frac{\eta}{\kappa}\bar{\boldsymbol{\Sigma}}_x^{\leq\kappa})^{(t-t_o^\kappa)}(\theta_{MT}(t_0^\kappa) - \boldsymbol{\theta}_{[1,\kappa]}^\star) \tag{13}$$

Note that applying Assumption 2 we directly have $\eta m\,\boldsymbol{I} \preccurlyeq \frac{\eta}{\kappa}\bar{\boldsymbol{\Sigma}}_x^{\leq\kappa} \preccurlyeq \eta M\,\boldsymbol{I}$, which allows us to use the same convergence statements for the ST and MT objectives.

The ST agent is reset after every task, and thus $\theta_{ST}(t_0^\kappa) = \boldsymbol{\theta}_0\,\forall\kappa$. In contrast, the MT agent is never reset and therefore it starts the new task from where it ended the last one $\theta_{MT}(t_0^\kappa) = \boldsymbol{\theta}_0 \iff t_0^\kappa = 0$. The task initialization $\theta_{MT}(t_0^\kappa)$ admits a closed-form expression:

$$\theta_{MT}(t_0^{\kappa+1}) = \sum_{j=0}^{\kappa} \Big[ \prod_{i=j+1}^{i} \underbrace{(\boldsymbol{I} - \frac{\eta}{i}\bar{\boldsymbol{\Sigma}}_x^{\leq i})^{h_i}}_{P_i} \Big] \left( \boldsymbol{I} - (\boldsymbol{I} - \frac{\eta}{j}\bar{\boldsymbol{\Sigma}}_x^{\leq j})^{h_j} \right) \boldsymbol{\theta}_{[1,j]}^\star$$

$$\tag{14}$$

$$= \sum_{j=0}^{\kappa} \Big[ \prod_{i=j+1}^{i} P_i{}^{h_i} \Big] \left( \boldsymbol{I} - P_j^{h_j} \right) \boldsymbol{\theta}_{[1,j]}^\star$$

where, with an abuse of notation we denote $P_0 = 0$ and $\boldsymbol{\theta}_{\leq 0}^\star = \boldsymbol{\theta}_0$.

## A.2 Average lifelong error difference

**Lemma 5.** *For any strictly convex loss $R$, i.e., there exists $m, M > 0$ such that $m\boldsymbol{I} \leq \nabla^2 R(\boldsymbol{\theta}) \leq M\boldsymbol{I}$ for all $\boldsymbol{\theta}$, the convergence of (full-batch) discrete time gradient descent with learning rate $\eta$ is geometric and we have:*

$$\|\theta(k) - \boldsymbol{\theta}^*\|_2 \leq (1 - \eta m)^k\|\boldsymbol{\theta}_0 - \boldsymbol{\theta}^*\|_2 \qquad R(\theta(k)) - R(\boldsymbol{\theta}^*) \leq (1 - \eta m)^{2k}\|\boldsymbol{\theta}_0 - \boldsymbol{\theta}^*\|_{\boldsymbol{\Sigma}_x}^2$$

$$\|\theta(k) - \boldsymbol{\theta}^*\|_2 \geq (1 - \eta M)^k\|\boldsymbol{\theta}_0 - \boldsymbol{\theta}^*\|_2 \qquad R(\theta(k)) - R(\boldsymbol{\theta}^*) \geq (1 - \eta M)^{2k}\|\boldsymbol{\theta}_0 - \boldsymbol{\theta}^*\|_{\boldsymbol{\Sigma}_x}^2$$

*where $\boldsymbol{\Sigma}_x = \nabla^2 R(\boldsymbol{\theta})$.*

**Assumption 6** (Learning rate). The learning rate is chosen such that gradient descent converges to a minimum. If Assumption 2 is satisfied this is simply: $\eta < \frac{1}{M}$.

**Definition 7** (Decomposition of $\Delta_T$). We identify three separate elements which contribute independently to the average lifelong error difference $\Delta_T$, namely:

$$\Delta_T^I = \sum_{i=1}^{K} \mathcal{R}_i(\boldsymbol{\theta}_{[1,i]}^\star) - \mathcal{R}_i(\boldsymbol{\theta}_i^\star) \tag{15}$$

$$\Delta_T^{MT} = \sum_{i=1}^{K} \left( \frac{1}{h} \sum_{t=t_0^i+1}^{ih} \mathcal{R}_i(\boldsymbol{\theta}_{MT}^{(i)}(t)) - \mathcal{R}_i(\boldsymbol{\theta}_{[1:i]}^\star) \right) \tag{16}$$

$$\Delta_T^{ST} = \sum_{i=1}^{K} \left( \frac{1}{h} \sum_{t=t_0^i+1}^{ih} \mathcal{R}_i(\boldsymbol{\theta}_{ST}^{(i)}(t)) - \mathcal{R}_i(\boldsymbol{\theta}_i^\star) \right) \tag{17}$$

Further,

$$\Delta_T = \frac{1}{K}\Delta_T^{ST} - \frac{1}{K}\Delta_T^{MT} - \frac{1}{K}\Delta_T^I \tag{18}$$

**Theorem 8** (General upper bound on $\Delta_T$). *For clarity in the notation, we fix $h_\kappa = h$ for all tasks, and denote $\epsilon_m = (1 - \eta m)^2$ where $\eta$ is the GD step size and $m$ is from Assumption 2. If Assumption 6 and Assumption 2 are satisfied then the difference in average lifelong error of the ST and MT agents with dynamics described by Equation (12) admits the following upper bound:*

$$\Delta_T \leq \frac{1}{K}\sum_{\kappa=1}^{K} \left( \frac{1}{h} \cdot \frac{1 - \epsilon_m^h}{1 - \epsilon_m}\|\boldsymbol{\theta}_0 - \boldsymbol{\theta}_\kappa^*\|_{\boldsymbol{\Sigma}_x^\kappa}^2 + O(1/N_\kappa) \right) - \frac{1}{K}\Delta_T^I \tag{19}$$

**Proof**   We start from the general decomposition of Definition 7. We bound each task term $\Delta_T^{ST}(t_0^\kappa + t), \Delta_T^{MT}(t_0^\kappa + t)$ separately. $\Delta_T^I \geq 0$ cannot be bounded further since it is not agent dependent. However we can rewrite is as follows:

$$\frac{1}{K}\Delta_T^I = \frac{1}{K}\sum_{\kappa=1}^{K}\left(\mathcal{R}_\kappa(\boldsymbol{\theta}_{[1,\kappa]}^\star) - \mathcal{R}_\kappa(\boldsymbol{\theta}_\kappa^\star)\right) = \frac{1}{K}\sum_{\kappa=1}^{K}\|\boldsymbol{\theta}_{[1,\kappa]}^\star - \boldsymbol{\theta}_\kappa^\star\|_{\boldsymbol{\Sigma}_x^\kappa}^2 \tag{20}$$

Both ST and MT are gradient descent agents that optimize a convex objective. By Lemma 5 and Assumption 2 we have that the train error with respect to the minimum will converge to $0$ at a geometric rate. Using a generic concentration argument to upper bound the difference between the empirical risk on the train set and the test error: $R_\kappa(\boldsymbol{\theta}_\kappa^*) - \mathcal{R}_\kappa(\boldsymbol{\theta}_\kappa^*)$ (the train and test set being identically distributed) we get:

$$\Delta_T^{ST}(t_0^\kappa + t) \leq (1 - \eta m)^{2t}\|\boldsymbol{\theta}_0 - \boldsymbol{\theta}_\kappa^*\|_{\boldsymbol{\Sigma}_x^\kappa}^2 + O(1/N_\kappa)$$

and

$$\Delta_T^{MT}(t_0^\kappa + t) \geq (1 - \eta M)^{2t}\|\boldsymbol{\theta}_{\mathrm{MT}}(t_0^\kappa) - \boldsymbol{\theta}_{[1:\kappa]}^\star\|_{\boldsymbol{\Sigma}_x^\kappa}^2 + O(1/N_\kappa)$$

First note that for $\kappa \gg 0$,

$$\Delta_T^{MT}(t_0^\kappa + t) \geq (1 - \eta M)^{2t}\|\boldsymbol{\theta}_{\mathrm{MT}}(t_0^\kappa) - \boldsymbol{\theta}_{[1:\kappa]}^\star\|_{\boldsymbol{\Sigma}_x^\kappa}^2 + O(1/N_\kappa) \gtrsim 0$$

because $\boldsymbol{\theta}_{\mathrm{MT}}(t_0^\kappa) \approx \boldsymbol{\theta}_{[1:\kappa-1]}^\star \approx \boldsymbol{\theta}_{[1:\kappa]}^\star$ is close to the minimum at the previous task, which is itself similar to the current minimum. So in general, we can grossly lower bound $\Delta_T^{MT}(t_0^\kappa + t) > 0$ without making a large error (see Lemma 9 for a formal proof).

Recognising that $(1 - \eta m)^{2t}$ forms a geometric series with base $\epsilon_m = (1 - \eta m)^2$, we can write :

$$\Delta_T \leq \frac{1}{K}\sum_{\kappa=1}^{K}\frac{1}{h}\left(\frac{1 - \epsilon_m^h}{1 - \epsilon_m}\|\boldsymbol{\theta}_0 - \boldsymbol{\theta}_\kappa^*\|_{\boldsymbol{\Sigma}_x^\kappa}^2\right) + O(1/N_\kappa) - \frac{1}{K}\Delta_T^I \tag{21}$$

This concludes the proof.

**Lemma 9.** *The error term due to the MT agent is negligible:*

$$\frac{1}{K}\sum_{\kappa=1}^{K}\|\boldsymbol{\theta}_{MT}(t_0^\kappa) - \boldsymbol{\theta}_{[1:\kappa]}^\star\|_{\boldsymbol{\Sigma}_x^\kappa}^2 \in o(1)$$

**Proof**   Using Equation (14) we can write

$$\begin{aligned}
\boldsymbol{\theta}_{\mathrm{MT}}(t_0^\kappa) - \boldsymbol{\theta}_{[1:\kappa]}^\star &= \sum_{j=0}^{\kappa}\big[\prod_{i=j+1}^{i}P_i^{h_i}\big]\left(\boldsymbol{I} - P_j^{h_j}\right)\boldsymbol{\theta}_{[1,j]}^\star - \boldsymbol{\theta}_{[1:\kappa]}^\star \\
&= \left(\boldsymbol{I} - P_{\kappa-1}^h\right)\boldsymbol{\theta}_{[1,\kappa-1]}^\star \\
&\quad + P_{\kappa-1}^h\left(\boldsymbol{I} - P_{\kappa-2}^h\right)\boldsymbol{\theta}_{[1,\kappa-2]}^\star \\
&\quad + P_{\kappa-1}^h P_{\kappa-2}^h\left(\boldsymbol{I} - P_{\kappa-3}^h\right)\boldsymbol{\theta}_{[1,\kappa-3]}^\star \\
&\quad + \ldots \\
&\quad + P_{\kappa-1}^h \ldots P_2^h\left(\boldsymbol{I} - P_1^h\right)\boldsymbol{\theta}_1^\star + P_{\kappa-1}^h \ldots P_1^h\boldsymbol{\theta}_0 - \boldsymbol{\theta}_{[1:\kappa]}^\star
\end{aligned}$$

By Assumption 2 we know $P_i^h \preccurlyeq (1 - \eta m)^h\boldsymbol{I}$ for all the tasks $i$ and thus we can ignore the contribution of all the terms $j < \kappa - 1$ in the norm:

$$\|\boldsymbol{\theta}_{\mathrm{MT}}(t_0^\kappa) - \boldsymbol{\theta}_{[1:\kappa]}^\star\|_{\boldsymbol{\Sigma}_x^\kappa}^2 \leq \|\boldsymbol{\theta}_{[1,\kappa-1]}^\star - \boldsymbol{\theta}_{[1:\kappa]}^\star\|_{\boldsymbol{\Sigma}_x^\kappa}^2$$

As $\kappa$ increases the average will converge to the final average $\boldsymbol{\theta}^{\star}_{[1,K]}$, and $\|\boldsymbol{\theta}^{\star}_{[1,\kappa-1]} - \boldsymbol{\theta}^{\star}_{[1:\kappa]}\|^2_{\boldsymbol{\Sigma}^{\kappa}_x} \to 0$. In general we can say that $\|\boldsymbol{\theta}_{\mathrm{MT}}(t^{\kappa}_0) - \boldsymbol{\theta}^{\star}_{[1:\kappa]}\|^2_{\boldsymbol{\Sigma}^{\kappa}_x}$ decreases with $\kappa$ and thus:

$$\frac{1}{K} \sum_{\kappa=1}^{K} \|\boldsymbol{\theta}_{\mathrm{MT}}(t^{\kappa}_0) - \boldsymbol{\theta}^{\star}_{[1:\kappa]}\|^2_{\boldsymbol{\Sigma}^{\kappa}_x} \in o(1)$$

**Corollary 10.** *Consider the setting where Assumption 2 and Assumption 6 are satisfied. If the instability of the sequence is null, i.e. $\Delta^I_T = 0$, then the upper bound in Theorem 8 is always positive.*

This result is a direct consequence of the general upper bound above. In particular, Lemma 9 shows that in such setting the error of the MT agents goes to 0 geometrically fast so it is the optimal type of agent.

**Theorem 11** (General lower bound on $\Delta_T$)**.** *In the same setting as Theorem 8, using $\epsilon_M = (1-\eta M)^2$, if Assumption 6 and Assumption 2 are satisfied then the difference in average lifelong error of the ST and MT agents with dynamics described by Equation (12) admits the following upper bound:*

$$\Delta_T \geq \frac{1}{K} \sum_{\kappa=1}^{K} \frac{1}{h} \left( \frac{1-\epsilon^h_M}{1-\epsilon_M} \|\boldsymbol{\theta}_0 - \boldsymbol{\theta}^*_{\kappa}\|^2_{\boldsymbol{\Sigma}^{\kappa}_x} - \frac{1-\epsilon^h_m}{1-\epsilon_m} \right) - \frac{1}{K} \Delta^I_T \tag{22}$$

**Proof**    The proof is similar to Theorem 11.

Again, we start from the general decomposition of Definition 7. Both ST and MT are gradient descent agents that optimize a convex objective. By Lemma 5 and Assumption 2 we have that the train error with respect to the minimum will converge to $0$ at a geometric rate. Using a generic concentration argument to upper bound the difference between the empirical risk on the train set and the test error: $R_{\kappa}(\boldsymbol{\theta}^*_{\kappa}) - \mathcal{R}_{\kappa}(\boldsymbol{\theta}^*_{\kappa})$ (the train and test set being identically distributed) we get:

$$\Delta^{ST}_T(t^{\kappa}_0 + t) \geq (1-\eta M)^{2t} \|\boldsymbol{\theta}_0 - \boldsymbol{\theta}^*_{\kappa}\|^2_{\boldsymbol{\Sigma}^{\kappa}_x} s$$

and

$$\Delta^{MT}_T(t^{\kappa}_0 + t) \leq (1-\eta m)^{2t} \|\boldsymbol{\theta}_{\mathrm{MT}}(t^{\kappa}_0) - \boldsymbol{\theta}^{\star}_{[1:\kappa]}\|^2_{\boldsymbol{\Sigma}^{\kappa}_x} + O(1/N_{\kappa})$$

Recognising that $(1-\eta m)^{2t}$ and $(1-\eta M)^{2t}$ form a geometric series with base $\epsilon_m = (1-\eta m)^2$ and $\epsilon_M = (1-\eta M)^2$ respectively, we can write :

$$\Delta_T \geq \frac{1}{K} \sum_{\kappa=1}^{K} \frac{1}{h} \left( \frac{1-\epsilon^h_M}{1-\epsilon_M} \|\boldsymbol{\theta}_0 - \boldsymbol{\theta}^*_{\kappa}\|^2_{\boldsymbol{\Sigma}^{\kappa}_x} - \frac{1-\epsilon^h_m}{1-\epsilon_m} \|\boldsymbol{\theta}_{MT}(t^{\kappa}_0) - \boldsymbol{\theta}^*_{\kappa}\|^2_{\boldsymbol{\Sigma}^{\kappa}_x} \right) + O(1/N_{\kappa}) - \frac{1}{K} \Delta^I_T$$

Applying Lemma 9 we know that the second term vanishes with $K$:

$$\frac{1}{K} \sum_{\kappa=1}^{K} \|\boldsymbol{\theta}_{MT}(t^{\kappa}_0) - \boldsymbol{\theta}^*_{\kappa}\|^2_{\boldsymbol{\Sigma}^{\kappa}_x} \in o(1)$$

and thus

$$\Delta_T \geq \frac{1}{K} \sum_{\kappa=1}^{K} \frac{1}{h} \left( \frac{1-\epsilon^h_M}{1-\epsilon_M} \|\boldsymbol{\theta}_0 - \boldsymbol{\theta}^*_{\kappa}\|^2_{\boldsymbol{\Sigma}^{\kappa}_x} - \frac{1-\epsilon^h_m}{1-\epsilon_m} \right) + O(1/N_{\kappa}) - \frac{1}{K} \Delta^I_T$$

This concludes the proof.

**Corollary 12.** *Consider the setting where Assumption 2 and Assumption 6 are satisfied. Let $V_K = \sum_{\kappa=1}^{K} \|\boldsymbol{\theta}_0 - \boldsymbol{\theta}_\kappa^*\|_{\boldsymbol{\Sigma}_x^\kappa}^2$ measure a quantity measuring the 'spread' of the task solution vectors, with respect to initialization, and further let $\omega_M = \frac{1-\epsilon_M^h}{1-\epsilon_M}$ and $\omega_m = \frac{1-\epsilon_m^h}{1-\epsilon_m}$. The lower bound in Theorem 11 is positive if the following is true:*

$$LB > 0 \iff V_K > \frac{\omega_m}{\omega_M} + \frac{h}{\omega_M}\Delta_T^I \tag{23}$$

*And thus if the instability of the sequence is null, i.e. $\Delta_T^I = 0$ then the lower bound in Theorem 11 is positive only if $V_K > \frac{\omega_m}{\omega_M}$.*

**Proof** Let $LB$ denote the lower bound on $\Delta_T$ of Theorem 11:

$$LB = \frac{1}{K}\sum_{\kappa=1}^{K}\frac{1}{h}\left(\frac{1-\epsilon_M^h}{1-\epsilon_M}\|\boldsymbol{\theta}_0 - \boldsymbol{\theta}_\kappa^*\|_{\boldsymbol{\Sigma}_x^\kappa}^2 - \frac{1-\epsilon_m^h}{1-\epsilon_m}\right) - \frac{1}{K}\Delta_T^I \tag{24}$$

$$LB > 0 \iff \frac{1}{K}\sum_{\kappa=1}^{K}\frac{1}{h}\left(\frac{1-\epsilon_M^h}{1-\epsilon_M}\|\boldsymbol{\theta}_0 - \boldsymbol{\theta}_\kappa^*\|_{\boldsymbol{\Sigma}_x^\kappa}^2\right) > \frac{1}{h}\frac{1-\epsilon_m^h}{1-\epsilon_m} + \frac{1}{K}\Delta_T^I \tag{25}$$

$$\frac{1-\epsilon_M^h}{1-\epsilon_M}\left(\frac{1}{K}\sum_{\kappa=1}^{K}\|\boldsymbol{\theta}_0 - \boldsymbol{\theta}_\kappa^*\|_{\boldsymbol{\Sigma}_x^\kappa}^2\right) > \frac{1-\epsilon_m^h}{1-\epsilon_m} + h\frac{1}{K}\Delta_T^I \tag{26}$$

$$\tag{27}$$

which concludes the proof.

Corollary 12 highlights the role of the task duration $h$ in the balance between ST and MT agents. As $h$ increases it becomes harder for the MT agent to match the performance of the ST agent. Another consequence of Corollary 12 is that a positive instability does not imply a positive $\Delta_T$. For instance, if the solutions are all $\delta$-close ($\delta = o(\frac{\omega_m}{\omega_M})$) to the initialization (e.g. by being of low norm) then the ST agent may still outperform the MT agent.

Moreover, since both the upper and lower bound on $\Delta_T$ vary as $h^{-1}$ we can say that $\Delta_T \in \Omega(h^{-1})$, which confirms that increasing the task duration will always lead to lower $\Delta_T$.

**Theorem 13** (Asymptotically tight bounds for $\Delta_T$). *Let $V_K = \sum_{\kappa=1}^{K}\|\boldsymbol{\theta}_0 - \boldsymbol{\theta}_\kappa^*\|_{\boldsymbol{\Sigma}_x^\kappa}^2$ as in Corollary 12. In the same setting as Theorems 8 and 11 if Assumption 2 and Assumption 6 are satisfied then the difference in average lifelong error described by Equation (12) can be tightly bounded as follows:*

$$\Delta_T \in \Theta\left(\frac{1}{K}\left(\frac{1}{h}V_K - \Delta_T^I\right) + \frac{1}{N_\kappa} + C\right) \tag{28}$$

*where $C$ is hiding a constant which depends only on the spectrum of the covariance matrices.*

**Proof** The theorem is a direct consequence of Theorem 8 and Theorem 11.

Interestingly, Theorem 13 highlights the nature of the dependence of $\Delta_T$ on $h$, which is essentially monotonic. The following corollary formalizes this observation.

**Corollary 3.** *In the same setting as Theorems 8 and 11 if Assumption 2 and Assumption 6 are satisfied then the difference in average lifelong error described by Equation (12) decreases monotonically with the task duration.*

Corollary 3 provides fundamental insight for our study, and has high practical relevance. The task duration $h$ is typically under the control of the agent designer. By Corollary 3 we know that increasing

the task duration will necessarily decrease the difference $\Delta_T$. However (Corollary 12) it is not granted that $\Delta_T$ will in general be negative, i.e. that a critical task duration exists in general.

In order to prove the existence of a critical task duration we need to consider the worst case scenario, i.e. the upper bound on $\Delta_T$. We are thus looking for cases where the instability is not 0, i.e. $\Delta_T^I > 0$. This is what the next set of results looks at.

**Theorem 14** (Negative $\Delta_T$ with positive instability). *Consider the setting where Assumption 2 and Assumption 6 are satisfied. If the instability of the sequence is strictly positive, then the upper bound in Theorem 8 is strictly negative if:*

$$h > \frac{\sum_{\kappa=1}^{K} \|\boldsymbol{\theta}_0 - \boldsymbol{\theta}_\kappa^*\|_{\boldsymbol{\Sigma}_x^\kappa}^2}{(1 - \epsilon_m) \sum_{\kappa=1}^{K} \|\boldsymbol{\theta}_{[1,\kappa]}^\star - \boldsymbol{\theta}_\kappa^\star\|_{\boldsymbol{\Sigma}_x^\kappa}^2} := \hat{h} \tag{29}$$

**Proof** We simply solve for $\Delta_T < 0$ in Theorem 8:

$$\Delta_T < 0 \Leftarrow h > \frac{\sum_{\kappa=1}^{K} \|\boldsymbol{\theta}_0 - \boldsymbol{\theta}_\kappa^*\|_{\boldsymbol{\Sigma}_x^\kappa}^2}{(1 - \epsilon_m) \sum_{\kappa=1}^{K} \|\boldsymbol{\theta}_{[1,\kappa]}^\star - \boldsymbol{\theta}_\kappa^\star\|_{\boldsymbol{\Sigma}_x^\kappa}^2}$$

**Theorem 15** (Existence of the critical task duration.). *In the setting where Assumption 2 and Assumption 6 are satisfied, if the instability of the sequence is strictly positive, gradient descent on the ST and MT convex objectives described in Section 4.2 gives rise to two parameter dynamics $\theta_{ST}(t)$ and $\theta_{MT}(t)$, such that there exists a finite critical task duration $\bar{h}$.*

**Proof** The result follows directly from Theorem 14. By definition (Proposition 1), the critical task duration is the minimal value of $h$ such that $\Delta_T < 0$. Since we know by Theorem 14 that $\Delta_T < 0 \; \forall h < \hat{h}$ then we know that $\bar{h} \leq \hat{h}$. Noticing that $\hat{h}$ is finite if the instability is strictly positive, then necessarily so is $\bar{h}$.

**Theorem 16** (Order of magnitude of the critical task duration). *With all the conditions of Theorem 13, ignoring the constants $C$ and $N_\kappa$ we know that the critical task duration admits the following asymptotic expression:*

$$\bar{h} \in \Theta\left(\frac{V_K}{\Delta_T^I}\right) \tag{30}$$

**Proof** Solving for $h$ in Theorem 13 and ignoring the terms depending on $N_\kappa$ or $C$ leads to the theorem statement.

Theorem 16 provides interesting insights. In particular, at higher instability in general the critical task duration is lower, which means that the multi-task solutions is more likely to achieve worse lifelong performance. At the same time, the norm of the solutions with respect to the initialization $V_K$ influences the balance between the two agents too. With the norm of the solutions tending to 0, the ST agent may still be more performing even in very stable environments.

A.3 TOY SETTINGS

For the toy settings in Figure 2 we can obtain explicit expressions by computing $\Delta_T^I$ and $\Delta_T^{ST}$ exactly.

In a one-dimensional problem the risk is simply $\mathcal{R}_i(\theta) = \sigma^2 (\theta - v_i)^2$, where w.l.o.g. we use $\Sigma_x = \sigma^2$. The MT objective minimizer after $\kappa$ tasks is:

$$\theta^\star_{[1,\kappa]} = (\sum_{i \leq \kappa} \hat{\Sigma}_x)^\dagger \underbrace{(\sum_{i \leq \kappa} \hat{\Sigma}_x \theta^\star_i)}_{\text{all average but the last one if odd}} \tag{31}$$

$$= (\kappa\sigma^2)^{-1}(\sigma^2 \left\lfloor \frac{\kappa}{2} \right\rfloor (\theta^\star_1 + \theta^\star_2) + \mathbf{1}_{\{\kappa \text{ odd}\}}\sigma^2\theta^\star_1) \tag{32}$$

$$= \begin{cases} \mu & \text{if } \kappa \text{ even} \\ \frac{\kappa-1}{\kappa}\mu + \frac{1}{\kappa}\theta^\star_1 & \text{if } \kappa \text{ odd} \end{cases} \tag{33}$$

where $\mu = \frac{1}{2}(\theta^\star_1 + \theta^\star_2)$ is the average solution. Thus, we can easily compute $\Delta^I_T$:

$$\sigma^2\|\theta^\star_{[1,\kappa]} - \theta^\star_\kappa\|^2 = \begin{cases} \frac{\sigma^2}{2}(\theta^\star_1 - \theta^\star_2)^2 & \text{if } \kappa \text{ even} \\ \frac{\sigma^2}{2} \cdot \frac{\kappa-1}{\kappa}(\theta^\star_1 - \theta^\star_2)^2 & \text{if } \kappa \text{ odd} \end{cases} \to_{\kappa \to \infty} \frac{\sigma^2}{2}(\theta^\star_1 - \theta^\star_2)^2 \tag{34}$$

$$\Delta^I_T = \sum_{k=1}^K \sigma^2\|\theta^\star_{[1,\kappa]} - \theta^\star_\kappa\|^2 = K\frac{\sigma^2}{2}(\theta^\star_1 - \theta^\star_2)^2 \tag{35}$$

Further, in Figure 2 we use $\theta_0 = 0$, therefore we have:

$$V_K = \sum_{\kappa=1}^K \sigma^2(\theta_0 - \theta^\star_\kappa)^2 = K\frac{\sigma^2}{2}(\theta^{\star 2}_1 + \theta^{\star 2}_2) \tag{36}$$

Finally by Theorem 14 we know that the critical task duration is at most:

$$\bar{h} \leq \frac{V_K}{\Delta^I_T} = \frac{\theta^{\star 2}_1 + \theta^{\star 2}_2}{(1 - \epsilon)(\theta^\star_1 - \theta^\star_2)^2} \tag{37}$$

where $\epsilon = (1 - \sigma\eta)^2$.

In our toy example in Figure 2, we chose $\eta = 0.01$ and $\sigma^2 = 9$, $\theta^\star_1 = 1$ and $\theta^\star_2 = 2$ in the left plot and $\theta^\star_2 = 1.1$ in the right plot. So we can solve for $\bar{h}^{left} \leq 29.09$ and $\bar{h}^{right} \leq 7478.9$.

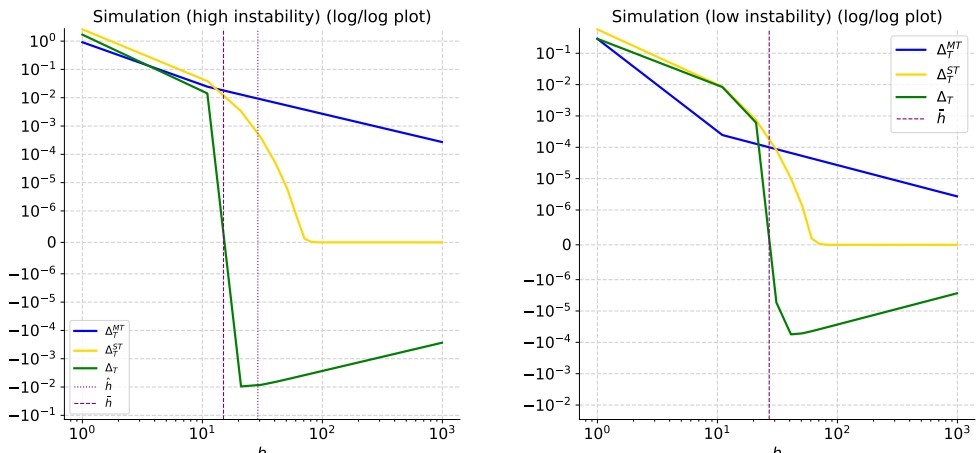

Figure 5: Simulation of $\Delta_T$ as a function of $T$ for the two toy settings of Figure 2.

**Simulations.** In order to get the precise value of $\bar{h}$ for our two toy settings we run 100 simulations as we vary $h \in [1, 1000]$, keeping all the other variables fixed. We plot the average lifelong error of ST and MT, and the respective $\Delta_T$ as a function of $T$ in Figure 5. The observed $\bar{h}$ is way lower

than its predicted upper bound $\hat{h}$, however the critical task duration is lower for higher instability -as expected. Also notice that when $\Delta_T^{ST} \approx 0$ the $\Delta_T$ grows less negative as $h$ is increased. This is a case that is not covered by the theory, since we work with the approximation $\frac{1}{K}\Delta_T^{MT} \approx 0$, whereas at very high $h$, the effect of $\frac{1}{K}\Delta_T^{MT}$ is much more pronounced compared to $\frac{1}{K}\Delta_T^{ST}$.

## A.4 OVERPARAMETRIZATION

Assumption 2 implies that the number of data points for each task $N_\kappa$ is at least equal to the number of parameters of the model $p$, i.e. $\min_\kappa N_\kappa \geq p$. If this condition is not satisfied, there exist infinitely many vectors which minimize the loss. It is known that gradient descent has an implicit bias towards minimum norm solutions (Gunasekar et al., 2018; Zhang & Sutton, 2020). Therefore, without changing the characteristics of the solution, we can augment the task loss with a regularizer. Denoting the overparametrized case with the $^o$ superscript:

$$\mathcal{R}_\kappa^o(\boldsymbol{\theta}) = \mathbb{E}_{x,\xi}\left[\langle \boldsymbol{\theta}_\kappa^\star - \boldsymbol{\theta}, \boldsymbol{x}\rangle]^2 + \lambda\|\boldsymbol{\theta}\|^2 \right. \tag{38}$$

$$= (\boldsymbol{\theta} - \boldsymbol{\theta}_\kappa^\star)^\top \boldsymbol{\Sigma}_x(\boldsymbol{\theta} - \boldsymbol{\theta}_\kappa^\star) + \lambda\|\boldsymbol{\theta}\|^2 \tag{39}$$

$$R_\kappa^o(\boldsymbol{\theta}) = \frac{1}{N_\kappa}\sum_{i=1}^{N_\kappa}\left[(\boldsymbol{\theta}_\kappa^\star - \boldsymbol{\theta})^\top \boldsymbol{x}_i\right]^2 + \lambda\|\boldsymbol{\theta}\|^2 \tag{40}$$

$$= (\boldsymbol{\theta} - \boldsymbol{\theta}_\kappa^\star)^\top \hat{\boldsymbol{\Sigma}}_x(\boldsymbol{\theta} - \boldsymbol{\theta}_\kappa^\star) + \lambda\|\boldsymbol{\theta}\|^2 \tag{41}$$

Next, we reproduce some key steps of our analysis with this modified loss in order to show that our derivations are readily extended to the overparametrized case.

The new minimizers of the ST and MT objectives are:

$$\boldsymbol{\theta}_\kappa^{o,\star} = \operatorname{argmin}_{\boldsymbol{\theta}} \Omega_{ST}(\boldsymbol{\theta}, \kappa) = (\lambda I + \boldsymbol{\Sigma}_x^\kappa)^{-1}\boldsymbol{\Sigma}_x^\kappa \boldsymbol{\theta}_\kappa^\star \tag{42}$$

$$\boldsymbol{\theta}_{[1,\kappa]}^{o,\star} = \operatorname{argmin}_{\boldsymbol{\theta}} \Omega_{MT}(\boldsymbol{\theta}, \kappa) = (\kappa\lambda I + \boldsymbol{\Sigma}_x^{\leq\kappa})^{-1}\bar{\boldsymbol{\theta}}_{[1,\kappa]}^\star \tag{43}$$

And the gradient descent dynamics for the two agents take the form:

$$\theta_{ST}^o(t) \leftarrow \theta_{ST}^o(t-1) - \eta\nabla_{\theta_{ST}^o(t-1)}R_\kappa^o(\boldsymbol{\theta})$$

$$= \theta_{ST}^o(t-1) - \eta\left(\hat{\boldsymbol{\Sigma}}_x^\kappa\left(\theta_{ST}^o(t-1) - \boldsymbol{\theta}_\kappa^\star\right) + \lambda\theta_{ST}^o(t-1)\right) \tag{44}$$

$$= (1 - \eta\lambda)\,\theta_{ST}^o(t-1) - \eta\,\hat{\boldsymbol{\Sigma}}_x^\kappa\left(\theta_{ST}(t-1) - \boldsymbol{\theta}_\kappa^\star\right)$$

$$\theta_{MT}^o(t) \leftarrow \theta_{MT}^o(t-1) - \frac{\eta}{\kappa}\sum_{i=1}^{\kappa}\nabla_{\theta_{MT}^o(t-1)}R_i^o(\boldsymbol{\theta})$$

$$= (1 - \eta\lambda)\,\theta_{MT}^o(t-1) - \frac{\eta}{\kappa}\sum_{i\leq\kappa}\hat{\boldsymbol{\Sigma}}_x^i\left(\theta_{MT}(t-1) - \boldsymbol{\theta}_i^\star\right) \tag{45}$$

Let $\lambda' = 1 - \eta\lambda$. Solving the recursion we have:

$$\theta_{ST}^o(t) = \boldsymbol{\theta}_\kappa^{o,\star} + (\lambda'\boldsymbol{I} - \eta\hat{\boldsymbol{\Sigma}}_x^\kappa)^{(t-t_o^\kappa)}\left(\boldsymbol{\theta}_0 - \boldsymbol{\theta}_\kappa^{o\star}\right) \tag{46}$$

$$\theta_{MT}^o(t) = \boldsymbol{\theta}_{[1,\kappa]}^{o,\star} + (\lambda'\boldsymbol{I} - \frac{\eta}{\kappa}\bar{\boldsymbol{\Sigma}}_x^{\leq\kappa})^{(t-t_o^\kappa)}\left(\theta_{MT}(t_0^\kappa) - \boldsymbol{\theta}_{[1,\kappa]}^{o,\star}\right) \tag{47}$$

We now propose an adapted version of Lemma 5, which crucially does not require the empirical covariance to be full rank, thus guaranteeing convergence in the overparametrized regime.

**Lemma 17** (Overparametrized convergence under regularization.)**.** *For any convex loss $R$ with added norm regularizer $\lambda\|\boldsymbol{\theta}\|^2$, such that $m\boldsymbol{I} \leq \nabla^2 R(\boldsymbol{\theta}) \leq M\boldsymbol{I}$ for $m, M \in \mathbb{R}^+$ and $0 < \lambda < \eta^{-1} - M$, the convergence of (full-batch) discrete time gradient descent with learning rate $\eta$ is geometric and*

*we have:*

$$\|\theta(k) - \boldsymbol{\theta}^*\|_2 \leq (1 - \eta m')^k \|\boldsymbol{\theta}_0 - \boldsymbol{\theta}^*\|_2 \qquad R(\theta(k)) - R(\boldsymbol{\theta}^*) \leq (1 - \eta m')^{2k} \|\boldsymbol{\theta}_0 - \boldsymbol{\theta}^*\|_{\boldsymbol{\Sigma}_x}^2$$

$$\|\theta(k) - \boldsymbol{\theta}^*\|_2 \geq (1 - \eta M')^k \|\boldsymbol{\theta}_0 - \boldsymbol{\theta}^*\|_2 \qquad R(\theta(k)) - R(\boldsymbol{\theta}^*) \geq (1 - \eta M')^{2k} \|\boldsymbol{\theta}_0 - \boldsymbol{\theta}^*\|_{\boldsymbol{\Sigma}_x}^2$$

*where $m' = m + \lambda$ and $M' = M + \lambda$, $\boldsymbol{\Sigma}_x = \nabla^2 R(\boldsymbol{\theta})$ and $\boldsymbol{\theta}^\star$ is the minimizer of the regularized objective.*

**Proof**    Let us consider the ST agent case, as the proof for the MT agent is similar. By Equation (46) we know that the GD estimate converges to the minimizer $\boldsymbol{\theta}_\kappa^{o,\star}$ exponentially fast:

$$\|\theta(k) - \boldsymbol{\theta}_\kappa^{o,\star}\|_2 \leq (1 - \eta m')^k \|\boldsymbol{\theta}_0 - \boldsymbol{\theta}_\kappa^{o,\star}\|_2$$

$$\|\theta(k) - \boldsymbol{\theta}_\kappa^{o,\star}\|_2 \geq (1 - \eta M')^k \|\boldsymbol{\theta}_0 - \boldsymbol{\theta}_\kappa^{o,\star}\|_2$$

The resulting estimation error is:

$$\mathcal{R}_\kappa(\theta(k)) = (\theta(k) - \boldsymbol{\theta}_\kappa^\star)^\top \boldsymbol{\Sigma}_x^\kappa (\theta(k) - \boldsymbol{\theta}_\kappa^\star) \tag{48}$$

$$= (\theta(k) - \boldsymbol{\theta}_\kappa^{o,\star})^\top \boldsymbol{\Sigma}_x^\kappa (\theta(k) - \boldsymbol{\theta}_\kappa^{o,\star}) + (\boldsymbol{\theta}_\kappa^{o,\star} - \boldsymbol{\theta}_\kappa^\star)^\top \boldsymbol{\Sigma}_x^\kappa (\boldsymbol{\theta}_\kappa^{o,\star} - \boldsymbol{\theta}_\kappa^\star) \tag{49}$$

$$\leq (1 - \eta m')^{2k} \|\boldsymbol{\theta}_0 - \boldsymbol{\theta}_\kappa^{o,\star}\|_{\boldsymbol{\Sigma}_x^\kappa}^2 + \|\boldsymbol{\theta}_\kappa^{o,\star} - \boldsymbol{\theta}_\kappa^\star\|_{\boldsymbol{\Sigma}_x^\kappa}^2 \tag{50}$$

What is left to prove is that $\|\boldsymbol{\theta}_\kappa^{o,\star} - \boldsymbol{\theta}_\kappa^\star\|_{\boldsymbol{\Sigma}_x^\kappa}^2 = 0$. We start by using the definition of $\boldsymbol{\theta}_\kappa^{o,\star}$:

$$\boldsymbol{\theta}_\kappa^{o,\star} - \boldsymbol{\theta}_\kappa^\star = (\lambda I + \boldsymbol{\Sigma}_x^\kappa)^{-1} \boldsymbol{\Sigma}_x^\kappa \boldsymbol{\theta}_\kappa^\star - \boldsymbol{\theta}_\kappa^\star \tag{51}$$

$$= ((\lambda I + \boldsymbol{\Sigma}_x^\kappa)^{-1} \boldsymbol{\Sigma}_x^\kappa - I) \boldsymbol{\theta}_\kappa^\star \tag{52}$$

Clearly, the difference is $0$ if the regularizer strength is $0$:

$$(\lambda I + \boldsymbol{\Sigma}_x^\kappa)^{-1} \boldsymbol{\Sigma}_x^\kappa - I = 0 \tag{53}$$

$$\iff \boldsymbol{\Sigma}_x^\kappa = \lambda I + \boldsymbol{\Sigma}_x^\kappa \tag{54}$$

$$\iff \lambda = 0 \tag{55}$$

In practice, $\lambda \to 0$ corresponds to the case where the population risk $R$ has a much stronger weight than the regularization strength in the objective (up to rescaling). Therefore, we may equivalently describe t$\boldsymbol{\theta}_\kappa^{o,\star}$ as the solution to the following constrained minimization problem:

$$\min \|\boldsymbol{\theta}\|^2 \qquad s.t. \ R_\kappa(\boldsymbol{\theta}) = 0 \tag{56}$$

Notice that this is the precise definition of the gradient descent solution in overparametrized settings.

In the overparametrized setting the condition $R_\kappa(\boldsymbol{\theta}) = 0$ is satisfied by any $\boldsymbol{\theta} = \boldsymbol{\theta}_\kappa' + \boldsymbol{P}_x \boldsymbol{v}$, where $\boldsymbol{v}$ is any vector in the parameter space, $\boldsymbol{P}_x$ is a projection operator on the orthogonal complement of the data space, i.e. $\boldsymbol{P}_x = \boldsymbol{I} - \boldsymbol{X}_\kappa^\dagger \boldsymbol{X}_\kappa$, and $\boldsymbol{\theta}_\kappa'$ is a solution to the task, i.e. $\boldsymbol{Y}_\kappa = \boldsymbol{\theta}_\kappa' \boldsymbol{X}_\kappa$. Thus picking $\boldsymbol{\theta}_\kappa^{o,\star} \in \{\boldsymbol{\theta} \,|\, \boldsymbol{\theta} = \boldsymbol{\theta}_\kappa' + \boldsymbol{P}_x \boldsymbol{v}\}$ necessarily $\|\boldsymbol{\theta}_\kappa^{o,\star} - \boldsymbol{\theta}_\kappa^\star\|_{\boldsymbol{\Sigma}_x^\kappa}^2 = 0$.

Lemma 17 bridges the regularized objective and the population risk, showing that convergence in one is necessarily linked to convergence in the other. Applying this lemma instead of Lemma 5, the results obtained in the underparametrized case can be extended to the overparametrized case without assumptions on the spectrum of the empirical covariance matrix.

## A.5   MEASURING THE INSTABILITY WITH THE NTK

A key takeaway of our theoretical analysis is that the optimal objective depends on the instability of the sequence. Thus, it is crucial to devise efficient and pragmatic, albeit precise, measures of instability. The two methods which we mention in Section 5.2 introduce noise in the estimate of $\Delta_T^I$ due to randomness in the optimization process, and in addition they both have high computational costs.

In what follows we explore a way to get rid of this noise using the Neural Tangent Kernel (NTK) (Jacot et al., 2018). The results are still in a preliminary form and thus they are not included in the main discussion, however they demonstrate potential in this direction of research.

Consider a linearization of the network around its initialization using the Neural Tangent Kernel (NTK) (Jacot et al., 2018):

$$f^{lin}(\boldsymbol{x}; \boldsymbol{\theta}_t) = f_0(\boldsymbol{x}) + \phi(\boldsymbol{x})^\top (\boldsymbol{\theta}_t - \boldsymbol{\theta}_0) \tag{57}$$

where $\phi(\boldsymbol{x}) = \partial_{\boldsymbol{\theta}_0} f_0(\boldsymbol{x})$ are the tangent kernel features. Minimizing a quadratic cost $R = \mathbb{E}_{(x,y)} [f^{lin}(x; \boldsymbol{\theta}) - y]^2$ averaged over a dataset $(\boldsymbol{X}, \boldsymbol{Y})$ in this new convex space we get the optimal weights:

$$\boldsymbol{\theta}^\star = \boldsymbol{\theta}_0 - \phi(\boldsymbol{X})^\top K(\boldsymbol{X}, \boldsymbol{X})^{-1} (f_0(\boldsymbol{X}) - \boldsymbol{Y}) \tag{58}$$

where $K(\boldsymbol{x}, \boldsymbol{x}') = \phi(\boldsymbol{x}) \phi(\boldsymbol{x}')^\top$ is the neural tangent kernel. In our continual learning setting, let $(\boldsymbol{X}_\kappa, \boldsymbol{Y}_\kappa)$ denote the dataset of task $\kappa$ and $(\boldsymbol{X}_{[1,\kappa]}, \boldsymbol{Y}_{[1,\kappa]})$ the concatenation of all the datasets $1, \ldots, \kappa$. Using Equation (58), and given a common initialization $\boldsymbol{\theta}_0$, the minimizers of the ST and MT objectives for task $\kappa$ are:

$$\boldsymbol{\theta}_\kappa^\star = \boldsymbol{\theta}_0 - \phi(\boldsymbol{X}_\kappa)^\top K(\boldsymbol{X}_\kappa, \boldsymbol{X}_\kappa)^{-1} (f_0(\boldsymbol{X}_\kappa) - \boldsymbol{Y}_\kappa) \tag{59}$$

$$\boldsymbol{\theta}_{[1,\kappa]}^\star = \boldsymbol{\theta}_0 - \phi(\boldsymbol{X}_{[1,\kappa]})^\top K(\boldsymbol{X}_{[1,\kappa]}, \boldsymbol{X}_{[1,\kappa]})^{-1} (f_0(\boldsymbol{X}_{[1,\kappa]}) - \boldsymbol{Y}_{[1,\kappa]}) \tag{60}$$

The instability is the average error of the MT minimizer compared to the average error of the ST minimizer. Suppose that the ST minimizer is optimal, i.e. that $y = f_0(\boldsymbol{x}) + \phi(\boldsymbol{x})^\top \boldsymbol{\theta}_\kappa^\star$, then we can measure the instability as the average error of the MT minimizer:

$$\Delta_T^I = \sum_{\kappa=1}^K \mathbb{E}_{(x,y)} \left[ f^{lin}(\boldsymbol{x}; \boldsymbol{\theta}_{[1,\kappa]}^\star) - y \right]^2$$

$$= \sum_{\kappa=1}^K \mathbb{E}_{(x,y)} \left[ \phi(\boldsymbol{x})^\top \boldsymbol{\theta}_{[1,\kappa]}^\star - \phi(\boldsymbol{x})^\top \boldsymbol{\theta}_\kappa^\star \right]^2$$

$$= \sum_{\kappa=1}^K (\boldsymbol{\theta}_{[1,\kappa]}^\star - \boldsymbol{\theta}_\kappa^\star)^\top \mathbb{E}_{\boldsymbol{x}} \left[ K(\boldsymbol{x}, \boldsymbol{x}) \right] (\boldsymbol{\theta}_{[1,\kappa]}^\star - \boldsymbol{\theta}_\kappa^\star)$$

$$= \sum_{\kappa=1}^K \|\boldsymbol{\theta}_{[1,\kappa]}^\star - \boldsymbol{\theta}_\kappa^\star\|_{\Sigma_{K_x}^\kappa}^2$$

where $\Sigma_{K_x}^\kappa = \mathbb{E}_{\boldsymbol{x}} \left[ K(\boldsymbol{x}, \boldsymbol{x}) \right]$ is the data covariance matrix in the kernel feature space. Let $\boldsymbol{\Xi}_\kappa = f_0(\boldsymbol{X}_\kappa) - \boldsymbol{Y}_\kappa$ denote the residuals at initialization. Then:

$$\boldsymbol{\theta}_{[1,\kappa]}^\star - \boldsymbol{\theta}_\kappa^\star = \phi(\boldsymbol{X}_\kappa)^\top K(\boldsymbol{X}_\kappa, \boldsymbol{X}_\kappa)^{-1} \boldsymbol{\Xi}_\kappa - \phi(\boldsymbol{X}_{[1,\kappa]})^\top K(\boldsymbol{X}_{[1,\kappa]}, \boldsymbol{X}_{[1,\kappa]})^{-1} \boldsymbol{\Xi}_{[1,\kappa]}$$

This quantity can be measured directly at initialization, and is exact in the infinite width limit, i.e. $\lim_{width \to \infty} \Delta_I^T \to \Delta_T^{I,\infty}$.

# B    REVIEW OF CONTINUAL LEARNING ALGORITHMS AND THE LINK TO THE MULTI-TASK OBJECTIVE

In this section we replicate some of the findings in the literature regarding the connection between existing CL algorithms and the multi-task objective. The discussion is mainly based on Yin et al. (2020) and Lanzillotta et al. (2024). We proceed by algorithm families, following the categorization of Parisi et al. (2019).

## B.1    REGULARIZATION METHODS

Let $\Omega_{CL}$ be the objective of a general CL algorithm. Yin et al. (2020) consider $\Omega_{CL}$ of the form:

$$\Omega_{CL}(\boldsymbol{\theta}, \kappa) = \frac{1}{\kappa} \sum_{i=1}^{\kappa} \hat{R}_i(\boldsymbol{\theta}) \tag{61}$$

where $\hat{R}_i(\boldsymbol{\theta})$ is an approximation of $R_i(\boldsymbol{\theta})$ based on a second order Taylor expansion centered at the task minimizer $\boldsymbol{\theta}_i^\star$. Thus in practice $\Omega_{CL}(\boldsymbol{\theta}, \kappa)$ approximated the MT objective $\Omega_{MT}(\boldsymbol{\theta}, \kappa)$. In Section 4 (Yin et al., 2020) it is shown how two popular regularization based methods implement $\Omega_{CL}$. We loosely follow their arguments here.

**Elastic Weight Consolidation.**    Kirkpatrick et al. (2017) use the approximation

$$\hat{R}_i(\boldsymbol{\theta}) = (\boldsymbol{\theta}_i^\star - \boldsymbol{\theta})^\top F_i (\boldsymbol{\theta}_i^\star - \boldsymbol{\theta})$$

where $F_i$ is the Fisher information matrix computed at $\boldsymbol{\theta}_i^\star$ (Equation 3, Kirkpatrick et al., 2017). For computational reasons, they approximate $F_i$ by zeroing the off diagonal entries. If the loss function is the negative log-likelihood, and we obtained the ground truth probabilistic model, then the Fisher information matrix is equivalent to the Hessian matrix, and $\hat{R}_i(\boldsymbol{\theta})$ coincides with the second order Taylor expansion when the gradient at $\boldsymbol{\theta}_i^\star$ is null.

**Kronecker factored Laplace approximation.**    Ritter et al. (2018) essentially refine the approximation of the Hessian matrix in EWC by considering a more sophisticated approximation of the fisher information matrix through a kronecker product rather than the diagonal approximation (Equations 5 and 9, Ritter et al., 2018).

**Synaptic Intelligence**    Zenke et al. (2017b) explicitly introduce an approximation of the task loss of the following form (Equation 4 and 6, Zenke et al., 2017b):

$$\hat{R}_i(\boldsymbol{\theta}) = R_i(\boldsymbol{\theta}_{old}) + (\boldsymbol{\theta}_{old} - \boldsymbol{\theta})^\top \boldsymbol{\Omega}_i (\boldsymbol{\theta}_{old} - \boldsymbol{\theta}) \tag{62}$$

where $\boldsymbol{\theta}_{old}$ is the value of the model parameters after training on the previous task and $\boldsymbol{\Omega}_i$ is a diagonal matrix which is an estimate of the parameter importance for the task $i$. In Section 4 (Zenke et al., 2017b) they demonstrate that under certain stability assumptions $\boldsymbol{\Omega}_i$ is directly related to the Hessian computed at $\boldsymbol{\theta}_{old}$. Thus also the SI method enters the general characterization of (Yin et al., 2020), with the difference that the Taylor approximation is not centered in $\boldsymbol{\theta}_i^\star$ but in $\boldsymbol{\theta}_{old}$. Lanzillotta et al. (2024) argue that this choice results in higher performance under long learning sequences.

In general, the conjecture proposed by Yin et al. (2020) is that many second order regularization based methods implicitly build an approximation of the form Equation (61) which is based on a second order Taylor expansion. A full review of the literature is out of the scope of this work and in general infeasible, without which the conjecture cannot be proven. Nonetheless, we believe this conjecture to be true for most existing regularization methods, and we do not make any claims on the ones which escape this characterization.

## B.2 Replay methods

Since Experience Replay was first introduced (Robins, 1995), several variants thereof have been proposed. In general, many replay-based algorithms optimize the same objective $\Omega_{CL}$ Equation (61), approximating the task loss $R_i$ through the use of a buffer:

$$\hat{R}_i(\boldsymbol{\theta}) = \sum_{(x,y) \in B_i} \ell(\boldsymbol{\theta}; x, y) \approx \sum_{(x,y) \in D_i} \ell(\boldsymbol{\theta}; x, y) \tag{63}$$

Importantly, often the samples from the buffer have an overall lower weight than the sample from the current task, e.g. by taking a gradient step on each. Thus, more accurately we say that many replay methods optimize the following objective:

$$\Omega_{rep}(\boldsymbol{\theta}, \kappa) = \frac{1}{\kappa} \sum_{i=1}^{\kappa} \alpha_i \hat{R}_i(\boldsymbol{\theta}) \tag{64}$$

where the task weight $\alpha_i$ is determined by the specific implementation of the algorithm. Our analysis of the MT objective can be easily extended to weighted average objectives, and we believe this conceptual framework to be an essential contribution of this work. In general, we demonstrate how to evaluate the optimality of any objective against a very simple baseline.

Next, we discuss other famous algorithms which belong to the replay category yet do not fall under the characterization of Equation (64). In doing so we mostly follow the arguments of Lanzillotta et al. (2024).

**Orthogonal Gradient Descent.** Orthogonal gradient descent (OGD) enforces orthogonality between the parameter update and the previous tasks output gradients (which are stored in the replay buffer). In order to see the connection to multi-task learning we must consider gradient-based updates. For an MT objective the gradients take the form:

$$\partial_{\boldsymbol{\theta}} \Omega_{MT}(\boldsymbol{\theta}, \kappa) = \frac{1}{\kappa} \sum_{i=1}^{\kappa} \partial_{\boldsymbol{\theta}} R_i(\boldsymbol{\theta}) \tag{65}$$

By a first order Taylor expansion, updating the parameters is the direction $-\partial_{\boldsymbol{\theta}} \Omega_{MT}(\boldsymbol{\theta}, \kappa)$ should decrease the objective by:

$$\Omega_{MT}(\boldsymbol{\theta}', \kappa) \approx \Omega_{MT}(\boldsymbol{\theta}, \kappa) - \eta \left\| \partial_{\boldsymbol{\theta}} \Omega_{MT}(\boldsymbol{\theta}, \kappa) \right\|^2 \tag{66}$$

The OGD condition enforcing orthogonality between the parameter update and the previous tasks output gradients instead modifies the MT loss as follows:

$$\Omega_{MT}(\boldsymbol{\theta}', \kappa) \approx \Omega_{MT}(\boldsymbol{\theta}, \kappa) - \eta \beta \| \frac{1}{\kappa} \partial_{\boldsymbol{\theta}} R_\kappa \|^2 \tag{67}$$

where $\beta = \cos(\partial_{\boldsymbol{\theta}} R_\kappa, \boldsymbol{\theta}' - \boldsymbol{\theta})$ is the angle between the projected update and the current task gradient -which must be non negative. Thus, the MT loss is still reduced by the OGD update, although the optimization is significantly slowed down (by a factor of $\sqrt{\kappa} \| \partial_{\boldsymbol{\theta}} \Omega_{MT} \|^2 / \beta \| \partial_{\boldsymbol{\theta}} R_\kappa \|^2$). Lanzillotta et al. (2024) prove that OGD implement an optimal quadratic constraint (Theorem 5.1, Lanzillotta et al., 2024), effectively minimizing the MT loss.

**Gradient Episodic Memory.** Gradient Episodic memory (GEM) minimizes a constrained objective where the parameter update has to be at a negative angle with the gradient of the previous task losses, i.e.:

$$\langle \partial_{\boldsymbol{\theta}} R_i, \boldsymbol{\theta}' - \boldsymbol{\theta} \rangle \leq 0 \tag{68}$$

The connection to the MT objective is similar to what we have seen for OGD. Simply considering a first order Taylor expansion of the MT objective we approximate its change due to the parameter update by:

$$\Omega_{MT}(\boldsymbol{\theta}', \kappa) \approx \Omega_{MT}(\boldsymbol{\theta}, \kappa) + \eta \sum_{i=1}^{\kappa} \beta_i \| \frac{1}{\kappa} \partial_{\boldsymbol{\theta}} R_i \|^2 \tag{69}$$

where $\beta_i = \langle \partial_{\boldsymbol{\theta}} R_i, \boldsymbol{\theta}' - \boldsymbol{\theta} \rangle$. Thus applying the GEM condition we know that the update reduces the MT objective.

### B.3 DYNAMIC ARCHITECTURE METHODS

Finally, we consider the set of dynamic architecture methods (e.g. Zhou et al., 2012; Rusu et al., 2016; Mallya & Lazebnik, 2018). Generally, these methods use new units or new parameters for each task, freezing the parameters where learning already happened. Effectively, one can formalize this considering a partition of the full set of parameters $S = \{\boldsymbol{\theta}_1, \ldots, \boldsymbol{\theta}_p\}$ in subsets $S_1, \ldots, S_K$ and enforcing the condition $(\boldsymbol{\theta}' - \boldsymbol{\theta})[S_i] = 0 \ \forall \, i \neq \kappa$ ($(\boldsymbol{\theta}' - \boldsymbol{\theta})$ is the vector of parameter update during task $\kappa$) and $\partial_{S_j} R_i(\boldsymbol{\theta}) = 0$ for all $j > i$ (Section 5, Lanzillotta et al., 2024).

To see the effect of this update strategy let's look at the angle of the update with the gradients of the MT objective:

$$\langle \, \boldsymbol{\theta}' - \boldsymbol{\theta}, \, \partial_{\boldsymbol{\theta}} \Omega_{MT}(\boldsymbol{\theta}, \kappa) \, \rangle = \frac{1}{\kappa} \sum_{i=1}^{\kappa} \langle \boldsymbol{\theta}' - \boldsymbol{\theta}, \partial_{\boldsymbol{\theta}} R_i(\boldsymbol{\theta}) \rangle \tag{70}$$

$$= \frac{1}{\kappa} \sum_{i=1}^{\kappa} \sum_{j=1}^{K} \langle (\boldsymbol{\theta}' - \boldsymbol{\theta})[S_j], \partial_{S_j} R_i(\boldsymbol{\theta}) \rangle \tag{71}$$

$$= \frac{1}{\kappa} \sum_{i=1}^{\kappa} \langle (\boldsymbol{\theta}' - \boldsymbol{\theta})[S_\kappa], \partial_{S_\kappa} R_i(\boldsymbol{\theta}) \rangle \quad \text{(first condition)} \tag{72}$$

$$= \frac{1}{\kappa} \langle (\boldsymbol{\theta}' - \boldsymbol{\theta})[S_\kappa], \partial_{S_\kappa} R_\kappa(\boldsymbol{\theta}) \rangle \quad \text{(second condition)} \tag{73}$$

The parameter update is typically a gradient-based update on the current loss (and satisfying the above conditions). Therefore we know that $\langle (\boldsymbol{\theta}' - \boldsymbol{\theta})[S_\kappa], \partial_{S_\kappa} R_\kappa(\boldsymbol{\theta}) \rangle < 0$ and thus in general $\langle \boldsymbol{\theta}' - \boldsymbol{\theta}, \partial_{\boldsymbol{\theta}} \Omega_{MT}(\boldsymbol{\theta}, \kappa) \rangle < 0$, which - by a first order Taylor expansion argument - results in a reduction in the MT objective.

Assuming that the optimization on each task is run to convergence, the final parameters at the end of each task are (local) minima of the task loss: $\boldsymbol{\theta}_\kappa^{end}[S_\kappa] = \arg\min_{\boldsymbol{\theta}[S_\kappa]} \{R_\kappa(\boldsymbol{\theta})\}$. Thus, after the entire sequence of tasks has been learned the model parameters $\boldsymbol{\theta}^{end}$ will satisfy the following conditions:

$$\begin{cases} \boldsymbol{\theta}^{end}[S_1] = \arg\min_{\boldsymbol{\theta}[S_1]} \{R_1(\boldsymbol{\theta})\} \\ \ldots \\ \boldsymbol{\theta}^{end}[S_K] = \arg\min_{\boldsymbol{\theta}[S_K]} \{R_K(\boldsymbol{\theta})\} \end{cases}$$

Thus effectively this class of methods assign a different subnetwork to each task, optimizing the tasks in isolation. Partitioning the network capacity compromises the performance on the task -which could be higher if the whole network were to be used- but it avoids forgetting.

Under a capacity constraint for each task, these methods minimize the MT loss, assuming the optimization converges to a minima for each task. To see why simply notice that $\min_{\boldsymbol{\theta} \in S} R_1(\boldsymbol{\theta}) + R_2(\boldsymbol{\theta}) \leq \min_{\boldsymbol{\theta} \in S} R_1(\boldsymbol{\theta}) + \min_{\boldsymbol{\theta} \in S} R_2(\boldsymbol{\theta})$.

## C    LIMITATIONS

Our work is a small step towards understanding and formalizing the existing assumptions in continual learning. The theoretical framework is limited to the convex case with linear models. Nevertheless we argue that theory is useful as long as it is predictive of behavior, even if it does not describe the actual setup.

Additionally, the proposed formalism is not descriptive enough to address complex shifts in the data distribution, as it relies on the assumption that there are contiguous time intervals (called tasks) where the data distribution is locally i.i.d..

Another limitation of the work is the choice of the MT agent, which is an abstract and unattainable rendition of continual learning algorithms. For example, experience replay may be biased to the current task, or simply fail to represent the past data distributions due to the limited buffer. In order to evaluate the exact degree of optimality of any specific algorithm the multitask objective should be modified in accordance with the algorithm.

Finally, the selective replay algorithm only provides a proof-of-concept idea of how the structure of the non-stationarity can be exploited by continual learning algorithms. In practice, one would need to estimate the sequence instability in order to run it. We believe that the online estimate of a sequence instability for the design of adaptive objectives is a promising avenue of future work.

## D    EMPIRICAL SETUP

### D.1    BENCHMARKS, NETWORKS AND GENERAL CONFIGURATION

Table 5: Supervised Learning benchmark statistics

| Benchmark | $K$ | Input Size | Classes | $N_\kappa$ |
|-----------|-----|------------|---------|------------|
| CLEAR | 10 | 224x224 | 100 | 109M |
| MD10 | 5 | 224x224 | 30 | $\in [1480, 27750]$ |
| PERMUTED CIFAR10 | 10 | 32x32 | 20 | 10K |

Table 6: MULTIDATASET datasets statistics

| Dataset | Classes | $N_\kappa$ |
|---------|---------|------------|
| StanfordCars | 196 | 1523 |
| FGVCAircraft | 100 | 2467 |
| DTD | 47 | 1480 |
| Food101 | 101 | 27750 |
| OxfordPet | 37 | 3680 |

#### D.1.1    CLEAR

The CLEAR dataset (Lin et al., 2021), a collection of images of 10 different classes spanning the years 2004-2014. We split the collection into 10 tasks, one for each year. The tasks are organised in their natural temporal ordering, i.e., by increasing year. All the input images are resized to 224x224 squares and normalised by subtracting the mean $\mu = [0.485, 0.456, 0.406]$ and dividing by $\Sigma = [0.229, 0.224, 0.225]$.

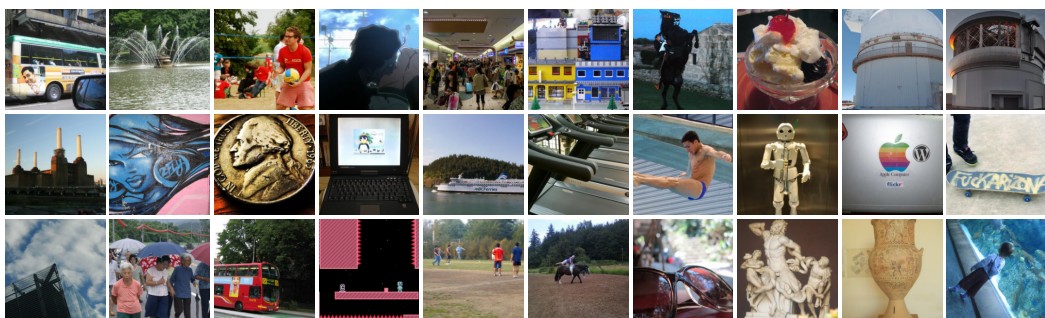

Figure 6: Samples from CLEAR benchmark. Each column corresponds to a different task.

### D.1.2   MULTIDATASET (MD5)

The MULTIDATASET benchmark consists of a sequence of 5 different open source classification datasets, with no semantic overlap between them. In particular, the tasks consists in classification of automobile models (Krause et al., 2013), aircraft models (Maji et al., 2013), textures (Cimpoi et al., 2014), dishes (Bossard et al., 2014) and pets (Parkhi et al.). Each dataset has originally a different number of classes, samples and a different input size - see Table 6. To avoid introducing biases in the models, we standardize all tasks to have only 30 classes, and we use the same batch size and amount of update steps in each task, regardless of the original dataset size. All the input images are resized to 224x224 squares and normalised by subtracting the mean $\mu = [0.485, 0.456, 0.406]$ and dividing by $\Sigma = [0.229, 0.224, 0.225]$. Additionally, the training dataset samples are augmented with random crops, random horizontal flips and random rotations of 15 degrees.

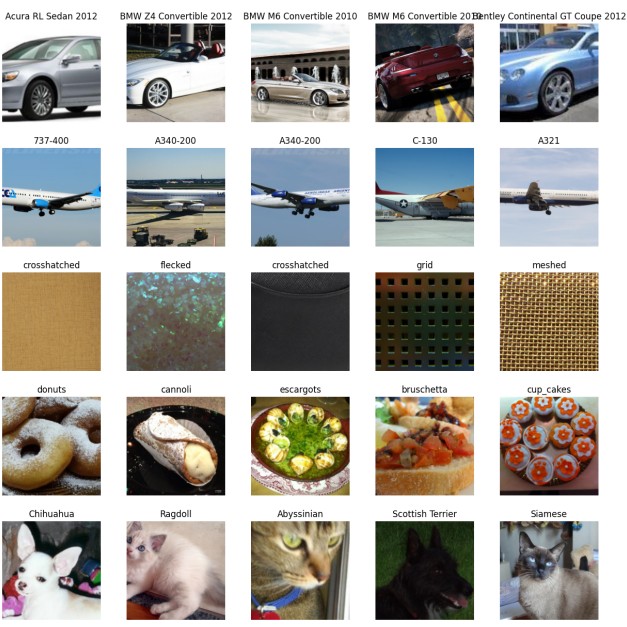

Figure 7: Samples from the MD5 benchmark. Each row corresponds to a different task.

### D.1.3 PERMUTED CIFAR 10

Permuted CIFAR 10 is a benchmark built from the CIFAR 10 dataset Krizhevsky & Hinton (2009), applying fixed random permutations to the images in the dataset. We use two different permutation sizes in all experiments, namely 16 and 32. The size of the permutation measures one length of the square box of pixels which will be permuted, centered at the center of the image (see Figure 8 and Figure 9 for examples). We refer to the respective benchmarks as 'CIFAR10 Permuted - 16' (PC-16) and 'CIFAR10 Permuted - 32' (PC-32). All the input images are normalised by subtracting the mean $\mu = [0.507, 0.486, 0.441]$ and dividing by $\Sigma = [0.267, 0.256, 0.276]$.

### D.1.4 META-WORLD

Meta-World is a collection of 50 distinct robotic manipulation tasks simulated in the MuJoCo physics engine. Each task involves controlling a robotic arm to interact with objects in its environment, such as pushing, picking, placing, opening drawers, or pressing buttons. The tasks are designed to test a range of skills and are suitable for evaluating both single-task and multi-task learning agents.

Each observation includes the robot's joint positions, velocities, and positions of relevant objects in the environment. For the multi-task agent, the observation is augmented with a task identifier.

Actions are continuous control signals sent to the robot's joints. Actions are sampled from a normal distribution defined by the policy network outputs. Log probabilities and entropies are computed to facilitate the learning process. Generalized Advantage Estimation (GAE) (Schulman et al., 2015) is utilized to compute advantages and target values for training.

### D.2 TRAINING PROCEDURES & NETWORKS

### D.2.1 SUPERVISED LEARNING EXPERIMENTS

All supervised learning agents consists of a network, an optimizer and a scheduler. In all supervised learning experiments the network is a residual network, *RN18* with the final linear head size being the number of classes in each task (100 for CLEAR, 30 for MD5 and 10 for PC). The final head is shared among all the tasks. See Table 10 and Table 9 for the network and optimization hyperparameters. The ST and MT agents are trained for the same number of steps $h$ with the same batch size per step.

*Single-Task Agent* Given a sequence of $K$ tasks the ST agent is trained to minimize a given loss function on the current task training data. The optimizer chosen is stochastic gradient descent. The ST agent network and optimizers are reset at the beginning of every task.

*Multi-Task Agent* Given a sequence of $K$ tasks the MT agent is trained to minimize a given loss function on the union of all the observed tasks training data, including the current task data. The optimizer chosen is stochastic gradient descent.

*Replay agents* In order to ensure comparability with the ST agent, the Experience Replay and Selective Replay agents are trained with the same batch size, which is equally partitioned between the current task data and the buffer data. The buffer is randomly filled at the end of each task with the data from the task. For all the agents we use a replay buffer of 500, meaning that we store 500 samples of each task in the buffer. While the ER agent is trained in a similar fashion as the MT agent, to minimize the loss on the the observed tasks, the SR agent ignores the buffer when the instability is high, i.e. in the second half of the sequence of tasks.

### D.2.2 REINFORCEMENT LEARNING EXPERIMENTS

In our reinforcement learning experiments, we aim to compare the performance of single-task and multi-task agents using Proximal Policy Optimization (PPO) Schulman et al. (2017) on the Meta-World benchmark Yu et al. (2020). PPO is a widely used policy-gradient method known for its stability and reliability in training deep reinforcement learning agents.

*Single-Task Agent.* For each task, we train a separate PPO agent with its own policy and value networks. The policy network is a multi-layer perceptron (MLP) consisting of two hidden layers with 128 units each and ReLU activation functions. The output layer produces the mean and standard

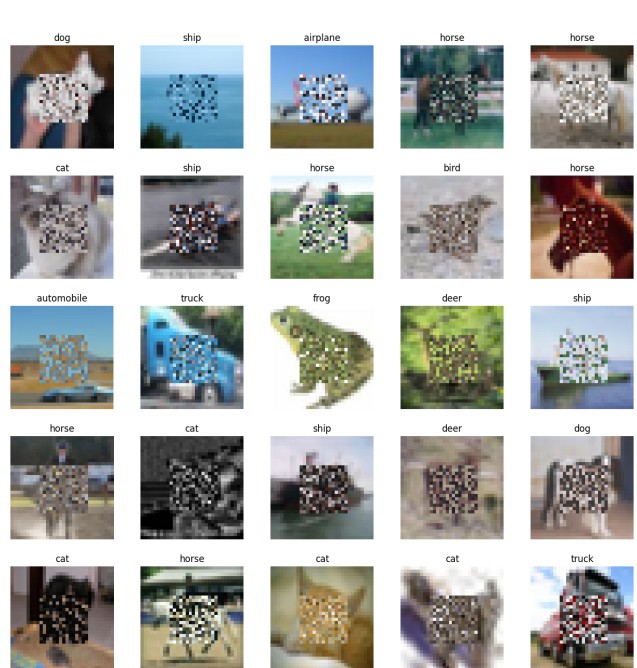

Figure 8: Samples from a Permuted CIFAR10 - 16 task.

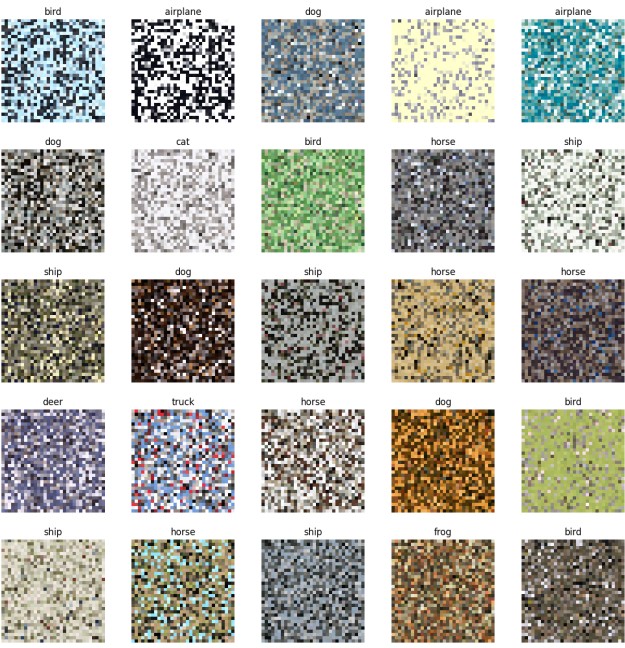

Figure 9: Samples from a Permuted CIFAR10 - 32 task.

deviation for a Gaussian action distribution. The value network shares the same architecture but outputs a scalar value estimate.

*Multi-Task Agent*. We train a single PPO agent across all selected tasks. The agent uses a shared policy network with the same architecture as the single-task agents. Multi-task agent uses a replay buffer to sample and update the PPO. The replay buffer at each task has the data from the current task and previous ones.

The RL agents were exposed to 10 tasks from ML10 benchmark, the tasks are as following: *Reach*, *Push*, *Pick & Place*, *Door Open*, *Drawer Close*, *Button Press*, *Peg Insert Side*, *Window Open*, *Sweep*, and *Basketball*. The order is preserved while running experiments for various task durations.

### D.2.3 HYPERPARAMETERS

### D.2.4 SUPERVISED LEARNING BENCHMARKS

The key hyperparameters which are tuned separately for each agent and benchmark are the learning rate, the batch size and the weight decay. The optimizer and scheduler are fixed across all supervised learning experiments. We employ SGD with a cosine annealing of the learning rate every $h$ time step, which means the learning rate is annealed over the course of each task and increased again at the beginning of the next task in order to allow the network to minimize the changing objective.

Table 7: Fixed HyperParameters for Supervised Learning Experiments. Note that the batch size has been tuned but the optimal batch size is the same for all agents and benchmarks

| HP | Value |
|---|---|
| **Momentum** | 0.9 |
| **Scheduler** | Cosine Annealing |
| **Batch Size** | 256 |
| **Optimizer** | SGD |

Table 8: Tuned HyperParameters for Supervised Learning Experiments

| Dataset | Agent | Network | LR | Weight Decay |
|---|---|---|---|---|
| CLEAR | ST | RN18 | 0.1 | $3 \times 10^{-4}$ |
| CLEAR | MT | RN18 | 0.1 | $1 \times 10^{-4}$ |
| C10 mixed | MT | RN18 | 0.075 | $7 \times 10^{-4}$ |
| C10 mixed | ST | RN18 | 0.055 | $8 \times 10^{-4}$ |
| PC-16 | MT | RN18 | 0.075 | $7 \times 10^{-4}$ |
| PC-16 | ST | RN18 | 0.055 | $8 \times 10^{-4}$ |
| PC-32 | MT | RN18 | 0.074 | $1 \times 10^{-3}$ |
| PC-32 | ST | RN18 | 0.06 | $1 \times 10^{-3}$ |
| MD5 | MT | RN18 | 0.08 | $6 \times 10^{-4}$ |
| MD5 | ST | RN18 | 0.089 | $9 \times 10^{-4}$ |

### D.2.5 ML10

We use the same set of hyperparameters for both agents where applicable to ensure a fair comparison. Key hyperparameters include a learning rate, a discount factor, and a clip ratio, enthropy coefficient, and lambda for GAE for training the PPO. Both agents are trained using the Adam optimizer. For the multi-task agent, gradients are calculated for each task and aggregated before the update step to ensure balanced learning across tasks. We train two type of agents, single-task and multi-task. The single task agent only receives observation from the current task, multi-task agent receives data from the current task as well as previous ones. We train these two type of agents for different task duration. For the RL experiments, we picked 50 and 500 episodes. All the results shown in tables Each episode is 500 time steps.

We use a batch size of 256 for updating the policy in ST agent and 512 for MT agent. In the multi-task setting, the batch is composed of an equal number of timesteps from each environment to prevent task imbalance.

Table 9: Optimal Configurations for ML10 for single-task and multi task agents

| Parameter | ST | MT |
|---|---|---|
| Batch Size | 256 | 512 |
| Entropy Coefficient | 0.02 | 0.02 |
| Learning Rate (Value) | $1 \times 10^{-3}$ | $1 \times 10^{-3}$ |
| Learning Rate (Policy) | $1 \times 10^{-4}$ | $1 \times 10^{-5}$ |
| Lambda for GAE | 0.95 | 0.8 |

Table 10: Software, hardware, and libraries used in the experiments

| | Python | MuJoCo | Meta-World | Gymnasium | GPU | RAM |
|---|---|---|---|---|---|---|
| Version | 3.8 | 2.3.2 | 2.0.0 | $\geq 1.0.0$ | *NVIDIA RTX 2080 Ti* | 128 GB |

### D.3 ADDITIONAL EMPIRICAL RESULTS

The table presents the task average reward for single-task and multi-task agents across various environments in the ML10 benchmark. Single-task agents generally perform better in most tasks, as seen in environments like *SawyerReachEnvV2* and *SawyerDrawerCloseEnvV2*. However, there are cases, such as *SawyerPushEnvV2*, where the multi-task agent outperforms the single-task agent.

Table 11: Task average reward over 500 episodes for single-task and multi-task agents in ML10.

| Task | reward$_{ST}$ ($\uparrow$) | reward$_{MT}$ ($\uparrow$) |
|---|---|---|
| *SawyerReachEnvV2* | $1.9130_{\pm 1.8771}$ | $1.8040_{\pm 1.6159}$ |
| *SawyerPushEnvV2* | $0.0349_{\pm 0.0456}$ | $0.0760_{\pm 0.3693}$ |
| *SawyerPickPlaceEnvV2* | $0.0079_{\pm 0.0055}$ | $0.0112_{\pm 0.0140}$ |
| *SawyerDoorEnvV2* | $0.5736_{\pm 0.2662}$ | $0.5052_{\pm 0.2829}$ |
| *SawyerDrawerCloseEnvV2* | $2.8774_{\pm 3.8648}$ | $2.2409_{\pm 3.4909}$ |
| *SawyerButtonPressTopdownEnvV2* | $0.4940_{\pm 0.4084}$ | $0.3761_{\pm 0.2232}$ |
| *SawyerPegInsertionSideEnvV2* | $0.0105_{\pm 0.0066}$ | $0.0124_{\pm 0.0088}$ |
| *SawyerWindowOpenEnvV2* | $0.4924_{\pm 0.4618}$ | $0.4294_{\pm 0.3024}$ |
| SawyerBasketballEnvV2 | $0.0114_{\pm 0.0082}$ | $0.0128_{\pm 0.0080}$ |

Figure 10 shows four heatmaps illustrating forward and backward transfer for single-task (left) and multi-task (right) agents across different environments. Each cell represents the amount of transfer

between pairs of tasks, with the x-axis indicating the source task and the y-axis indicating the target task. *Forward Transfer* measures how learning a previous task improves (or degrades) performance in a future task. Higher values indicate a positive impact, where experience from one task helps improve performance in another. The ST agent (a) shows strong forward transfer in a few pairs (e.g., *SawyerPickPlaceEnvV2* to *SawyerDrawerCloseEnv2*), while the MT agent (b) exhibits more consistent transfer patterns across several tasks. *Backward Transfer* measures the impact of learning a new task on previously learned ones. The ST agent suffers from low backward transfer while MT shows less severe negative transfer, suggesting better robustness when incorporating new tasks.

The forward transfer matrix is represented as an *upper triangular matrix*, this structure means that the matrix entries below the diagonal are zeros (or not applicable), while entries above the diagonal capture the influence of each task on tasks that are learned afterward. The backward transfer is represented as a *lower triangular matrix*, meaning that the entries above the diagonal are zeros, while entries below the diagonal capture the influence of learning a new task on earlier ones.

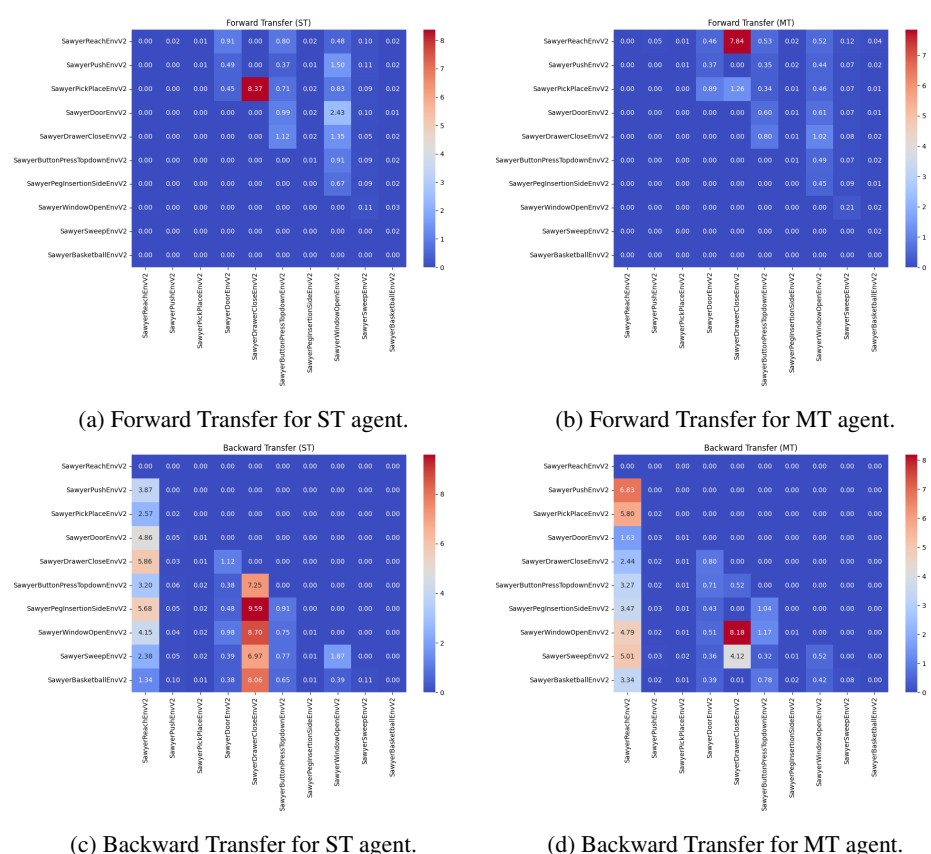

(a) Forward Transfer for ST agent.

(b) Forward Transfer for MT agent.

(c) Backward Transfer for ST agent.

(d) Backward Transfer for MT agent.

Figure 10: Forward and Backward Transfer matrices in ML10.

