# OpenReview forum: "Is multitask learning all you need in continual learning?"
_ICLR.cc/2025/Conference — Submitted to ICLR 2025_

### Official Review · Reviewer_Dyh5 · 2024-10-22

**Soundness:** 3
**Presentation:** 3
**Contribution:** 1
**Rating:** 5
**Confidence:** 4

**Summary:**

The paper defines "average life long error" as the average error the continual model had on the task it was trained on. Then, the paper compares between single-task and multi-task agents from that perspective, showing that either agent can achieve lower error depending on the specific task sequence. The paper proves that in a convex under-parameterized case, when training the agents on each task for a long enough period of time, the single-task agent actually reach better results. The paper continues with an empirical study, showing similar results in a range of simple toy experiments.

**Strengths:**

Originality: The paper introduces a novel metric, which allows for original analyses. The analyses give birth to elements like "instability", which can help in giving intuition to the difficulty of task sequences.

Clarity: The paper is written in a clear way, and it is easy to follow the presented ideas. The ideas presented are simple and intuitive, and the motivation behind them is generally clear.

Significance: Studying the reliance of continual learning algorithms on multitasks solutions is interesting and with a value to the community. The paper does make you question this point, which is not often addressed in the literature.

**Weaknesses:**

The "average lifelong error" sums the empirical risks of models trained sequentially, but only on the test data that comes from the distribution of the task that the models are trained on at that time. This effectively ignores the "continual" aspect of the models: in continual learning, we want to train a model first on one task, then on a second task, and expect the model to perform well on both tasks simultaneously.

Using the "average lifelong error" metric places multi-task agents at a disadvantage. While single-task models are evaluated solely on the data they are trained on, multi-task agents are also trained on unrelated data, which is not part of their evaluation. Consequently, unless the additional data seen by the multi-task model is highly correlated with the task at hand (as in the example shown in Figure 2b), it is expected that the single-task model will train faster, converge more efficiently, and achieve a better "average lifelong error" compared to the multi-task model.

Although this outcome seems intuitive given the experimental setup, the paper overemphasizes this result, as it claims that so far the multi-task agent served only as an upper bound, and this shows that it is not always true. Therefore, the manuscript asserts that "continual learning solutions should not and do not have to adhere to multitask objectives". I find this conclusion overstated: The metric is built in such a way that is skewed against multi-task learning, which is bound to fail it given the suggested metric, which does not capture the essence of continual learning. In all the experiments in the paper, when testing the total accuracy across all tasks, the multi-task agent does overperform the single-task agent.

That said, the proposed "average lifelong error" does provide valuable insight. Specifically, if a multi-task agent performs better than a single-task agent, it indicates that the unrelated data encountered by the multi-task agent is beneficial for training, and the method can serve as a useful measure of data similarity. The authors acknowledge this in the paper, but I believe this point should be emphasized more, while the broader claims about continual learning should be moderated. As a result, the quality of the paper drops (as the claims are not fully supported), and the significance of the findings is much more limited (as this point is much weaker than the general claims the paper suggests).

Nevertheless, I appreciate the soundness and depth of the analyses presented. Including both theoretical and empirical sections strengthens the work, even if the experiments are conducted on small-scale toy problems. However, given that the analysis focuses predominantly on the "average lifelong error," the insights regarding continual learning are somewhat limited.

**Questions:**

What is the motivation behind ignoring the risk over the previous task in the "average lifelong error"? In which situations would we want models that perform well throughout the entire learning, ignoring the final performance? If the performance on past tasks is forgotten, what intuition can be gained for continual learning, using this measurement?

---

> ### Author Response · Authors · 2024-11-19
>
> We thank the reviewer for the opportunity to clarify the core idea of the paper and its philosophical standpoint. Prompted by this review, we have revised the introduction of the paper, insisting on the motivation behind our specific choices. We believe that these changes have greatly improved the clarity of the paper.
>
> The reviewer correctly points out that "in continual learning, we want to train a model first on one task, then on a second task, and expect the model to perform well on both tasks simultaneously." This is what we mean when we say that continual learning algorithms implicitly aim at optimizing a multi-task objective ('performing well on all tasks' is another way to describe the MT objective). However, the question that our paper is posing to the continual learning community is whether this multi-task objective in practice achieves the end goals of the community. A good follow-up question is "what are the goals of the continual learning community?" In the paper, we follow recent literature ([1], [2], [3]) in recognizing that the ultimate goal of a continual learning agent is to *perform well throughout its lifetime*. When a system is deployed in the real world, it should leverage its previous knowledge to enhance its current performance, e.g., by learning faster, performing better, and retaining useful knowledge as long as it is beneficial. In this sense, remembering serves the purpose of enhancing current and future performance. Kumar et al. ([2]) have formalized this statement using information theory, showing theoretically that forgetting of 'useless' knowledge is not a bad thing when one cares about lifelong performance.
>
> Therefore, it is true that "Using the 'average lifelong error' metric places multi-task agents at a disadvantage" but -as you correctly notice- this depends on the correlation between the tasks, a concept which we formalize with our definition of "instability". This forms the main claim of our paper: that multi-task learning is not always optimal for lifelong performance. However, single-task learning (i.e., always erasing memory) is not always optimal either.
>
> Perhaps the most exciting consequence of this statement is that the design of optimal CL agents (where optimality is defined based on lifelong performance) requires objectives which are data dependent. In other words, if we want to build better CL agents we should develop methods to estimate the optimal amount of forgetting for a given sequence of tasks, which is a clear direction for future research.
>
> We acknowledge that the main claim may appear simple and overemphasized. However, we believe it has important implications for the field, and developing the experimental and theoretical framework to support the claim was not trivial.
>
> Overall, we understand that the reviewer may disagree with us on a philosophical level, as to what is supposed to be the main goal of continual learning. And rightfully so: this is an open question and a topic of ongoing debate within the CL community. However, this paper is not trying to argue about the "true" goal of continual learning, but rather, it takes a specific stand in this philosophical debate as a starting point. We hope the reviewer can appreciate the solidity of our work beyond the philosophical dispute.
>
> If we have answered the reviewer's question we kindly invite the reviewer to briefly check the introduction in the new version and provide feedback regarding the clarity of our motivation and positioning within the continual learning community. This aspect of our work is really essential to understand its value and so we would like to make sure that the message is conveyed correctly.
>
> [1] Hadsell, Raia, et al. "Embracing change: Continual learning in deep neural networks." Trends in cognitive sciences 24.12 (2020): 1028-1040.
>
> [2] Kumar, Saurabh, et al. "Continual learning as computationally constrained reinforcement learning." arXiv preprint arXiv:2307.04345 (2023).
>
> [3] Mundt, Martin, et al. "A wholistic view of continual learning with deep neural networks: Forgotten lessons and the bridge to active and open world learning." Neural Networks 160 (2023): 306-336.

---

> > ### Comment · Reviewer_Dyh5 · 2024-11-26
> >
> > I appreciate the many changes the authors incorporated into this new revision. The presentation has significantly improved, as many of the claims have been appropriately toned down, and the focus of the paper is much clearer.
> >
> > However, I still have some reservations regarding the current manuscript.
> >
> > Ultimately, continual learning is composed of both *forward transfer* - the ability to perform well on future tasks given past experience - and *backward transfer* - the ability to perform well on previous tasks given the information the network has learned.
> >
> > In the original manuscript, the claims of the paper seemed to address both of these aspects, suggesting that the multitask objective is fundamentally not well-suited for continual learning. The revised manuscript now makes a clearer distinction and focuses primarily on forward transfer, arguing that the multitask objective is not optimal in this specific regard. The authors demonstrate that there are datasets where an agent optimizing a single task at a time can outperform an agent optimizing all tasks simultaneously.
> >
> > While this narrower claim is more reasonable, it is also less impactful. Ultimately, the paper primarily shows that having additional out-of-distribution training data can be beneficial or detrimental depending on the degree of similarity between the data distributions. This is a well-known observation in the machine learning community and serves as the underlying rationale for why techniques such as pre-training and training on auxiliary data are effective in practice.
> >
> > Moreover, I find the paper’s framing within the continual learning domain somewhat unclear. Forward transfer is traditionally the main focus of online learning, where the goal is to improve performance on future tasks. Continual learning, on the other hand, encompasses both forward and backward transfer, aiming to balance the tradeoffs between them. While the authors have added connections to online learning in the new manuscript, the primary framing of the paper remains in the context of continual learning, which feels somewhat misaligned. This focus on forward transfer alone, while valid, seems better suited to the online learning domain, where backward transfer is not a concern.
> >
> > Additionally, I find that the claims of the paper remain too strong. The results suggest that the multitask solution is not optimal for forward transfer in some cases, but the manuscript appears to extrapolate this to argue that continual learning should move away from multitask solutions entirely. However, as continual learning is inherently a balance between forward and backward transfer, the findings instead seem to highlight the existence of a tradeoff between these objectives - a concept that has already been explored in previous works. Future solutions in continual learning likely need to integrate both multitask objectives and forward transfer considerations rather than abandoning multitask approaches altogether.
> >
> > A minor but important note: the current manuscript exceeds the 10-page limit and does not conform to the conference template. While I have not changed my score based on this issue,  as I hope this is relatively easy to fix. However, this must be fixed in any future revision.
> >
> > The revised manuscript has successfully addressed many of my initial concerns, resulting in a clearer and more focused presentation. However, given the remaining issues and the reduced significance of the revised claims, I am still inclined to recommend the rejection of the paper in its current form. I believe the work could benefit from further refinement and submission to a future venue. To reflect the improvements made, I am raising my score to 5.

---

> ### Author Response · Authors · 2024-11-27
>
> We thank the reviewer for their thoughtful response and active engagement in the rebuttal process. The points raised are certainly valid, and we find this discussion both stimulating and highly relevant to the community.
> We understand your reservations, particularly regarding the impact of our claims and the framing within the continual learning domain. Below, we would like to address your concerns in more detail:
>
> As noted by the reviewer, we have revised the manuscript to emphasize our focus on forward transfer rather than backward transfer, as this is central to our study. This focus naturally brings us closer to online learning, which traditionally emphasizes forward transfer.  The core of this discussion is whether our work is more aligned with continual learning or better categorized as online learning. We believe this is largely a philosophical question that ultimately leads to differing interpretations of the goals of continual learning.
> There are two main perspectives in this debate: a traditionally dominant view, which defines continual learning as the challenge of balancing forward and backward transfer, and a more recent perspective, which sees continual learning as the ability to perform well throughout the lifetime of the agent. In this work, we adopt the latter view, which aligns continual learning with fields like online learning and reinforcement learning. For example, Kumar et al [1]. frame continual learning as "computationally constrained reinforcement learning". We view this alignment with other fields positively, as the exchange of ideas and tools across disciplines can accelerate progress. One implicit goal of our work is indeed to bridge the gap between online learning and continual learning.
> Nonetheless, key differences remain: historically, online learning has been more theoretical, often relying on strong assumptions such as convexity, whereas continual learning has focused more on deep learning with an emphasis on practical applications.
>
> The reviewer mentioned that focus on forward transfer alone makes our findings less impactful, given that the benefits or detriments of out-of-distribution data are well-understood in the community. We recognize the validity of this observation. However, our intention is to clarify specific scenarios in which multitask objectives may lead to suboptimal forward transfer performance.
> For example, the concept of instability introduced with our theoretical analysis, allows for a quantitative assessment of the suboptimality of multitask objectives in a given environment.  We believe that highlighting these nuances, while not groundbreaking, offers a concrete basis for the development of *provably optimal objectives* for lifelong performance.
>
> Additionally, as summarized in the Discussion and Conclusion section, our argument is that "reliance on the multitask objective is not necessary" and that the optimal approach should depend on the data. This is not equivalent to "abandoning multitask approaches altogether"; rather, a multitask objective should be employed when it is optimal and set aside when it is not. More broadly, we envision that future research in continual learning should focus on defining data-dependent objectives that dynamically determine the appropriate balance of memory for the agent.
> Finally, given the philosophical nature of this discussion, unless the reviewer finds technical flaws in our work, we do not see a substantial basis for recommending rejection.
>
> [1] Kumar, S., Marklund, H., Rao, A., Zhu, Y., Jeon, H. J., Liu, Y., & Van Roy, B. (2021). Continual Learning as Computationally Constrained Reinforcement Learning. Advances in Neural Information Processing Systems (NeurIPS).

---

### Official Review · Reviewer_cvzj · 2024-10-28

**Soundness:** 3
**Presentation:** 3
**Contribution:** 3
**Rating:** 6
**Confidence:** 4

**Summary:**

This paper challenges the assumption that multi-task training is the upper bound for continual learning solutions that learn on a stream of such tasks. They formalize the limitations of multitask objectives and show cases where the multitask optimal solution is different from the continual optimal solution, both with a toy example and with a modified version of a "real-world" dataset.

**Strengths:**

- The authors propose to challenge an assumption that is often made in continual learning, bringing that topic into discussion. Challenging this assumption is reasonable and needs to be backed up preferably by both simple examples demonstrating why the assumption does not hold in some cases, and empirical results showing that this assumption can also be challenged in real world problems. The paper follows that structure which is good.
- The paper is well written and quite easy to follow.
- Many benchmarks are chosen for empirical evaluation

**Weaknesses:**

- **W1**: The toy setting is too simple and does not accurately reflect what could occur in the wild. This setting assumes something similar to a label switch while keeping the same head, which means that for instance an object would have to be recognized as something and then as something else, leading to an impossible solution for the multitask loss (even in the offline learning case this loss could not learn anything but overfit since it would be asked to learn from contradictory signals). I think a better example should show that even if there is a satisfying solution for multitask loss, learning using this loss does not result in an upper bound for the CL method (at least in term of learning efficiency)
- **W2**: As it is right now it is not clear why you split the empirical evaluation in two parts, one part using the CLEAR dataset and MD5 and another part using the CIFAR10 permuted. This needs to be explained more in details why these two parts are needed and what do you want to show in each part.
- **W3**: The main weakness of the paper in my opinion is that it is too disconnected from the rest of the online continual learning works. It is true that is online learning most metrics that you present are used and people care more about the rapidity of adaptation rather than the retaining of previous knowledge. But in most online continual learning  works, that is not the case, and metrics such as average accuracy, average anytime accuracy (area under the curve of AA) are used way more. So you need to justify why you focus only on these metrics. So far the only continual learning paper I know that used these is a paper on the CLOC dataset, where it is kind of justified to look at the adaptation metric since the knowledge needs to be "updated" and there is no need to "retain" previous knowledge. But in many of the benchmarks that you present, the retaining of previous knowledge is key to performing on the test set. So it is unfair to present the multitask baseline as "under performing" compared to the CL solution under these metrics that the MT baseline is not suppose to be the upper bound of. **In Short** , I agree that these metrics are important, but more classical metrics should also be reported and the advantage of CL methods (could be CL methods that use infinite memory) on these metrics should also be shown.
- **W4**: You claim in the paper that most CL methods use the multitask baseline as an upper bound, which I agree on. However, you also claim that most CL methods use the ERM objective. I think this is not entirely true. First of all, when there is no replay it is not the case most of the time. Secondly, even when replay is used, in general it is not used to get a precise estimation of the ERM objective because of the different weightings applied to the memory batch and current batch. Most CL methods would just draw one batch from the current task and one batch from  the memory and sum their loss to get the training loss. This results in giving more weight to the current task data compared to the memory when num_task > 2,  and it results in a different objective than the ERM. I think this part of the story could be tuned down a bit.

**Questions:**

- In the CIFAR10 benchmark used for empirical evaluation, you say that you vary the permutation size of the two datasets, but it is not explained what entries are permuted, do you chose a random pixel set to apply the permutation on ?
- I think there is a mistake in Figure 1, caption should say "On the right, ... while on the left, reverse is true"

---

> ### Author Response · Authors · 2024-11-19
>
> Thank you for your detailed and insightful comments. We appreciate the opportunity to address your concerns and provide further clarifications.
>
> 1. We have improved our toy settings significantly based on your feedback. The revised toy setting now provides a clearer and more nuanced demonstration of the key factors identified by our theoretical analysis. Specifically, we refined the mathematical formulation in the appendix (Section A.3) to better reflect more realistic scenarios where a multitask loss solution is applicable, addressing your valid concern about contradictory signals. This setting demonstrates that, even when a multitask loss is viable, it does not guarantee optimal performance.
>
> 2. We have clarified the structure of the empirical evaluation section. As per your suggestion, we added a description at the beginning of the section to explain the motivation behind splitting the experiments into two parts: the CLEAR dataset and MD5 vs. CIFAR10 Permuted. Specifically: The CLEAR dataset, MD5 and ML10 experiments focus on validating our main claim (that the MT objective is not always better) in realistic benchmarks which are used in practice.
>
>    The CIFAR10 Permuted experiments are meant to validate more specific and nuanced predictions from our theory, which can only be performed in environments where we can control the instability $\Delta_T^I$. Since permutations of the input significantly increase the learning task difficulty, we opted for a simpler task such as CIFAR10.
>
>    This restructuring helps clarify the rationale for including both parts and the insights they provide.
>
> 3. Regarding the connection to the online CL literature, we acknowledge that there was an important part of related work missing from the discussion. Prompted by your observation, we have added a paragraph (in Section 2) regarding the OCL literature and the connection between existing metrics and our average lifelong error. In particular, the so called 'average anytime accuracy' (as defined by [1]) should be equivalent to $1 - $ our average lifelong learning error, and it also corresponds to the online average accuracy (defined by [2]). Additionally, we reiterate that the average lifelong error is directly linked to another crucial metric in online learning -the dynamic regret.
>
>     In order to make our empirical results more easily accessible to the rest of the community, we have modified the experimental sections and we now report accuracies instead of errors.
>
> 4. We have clarified and softened the language around the claim that "most CL methods use the ERM objective."
> We acknowledge that the claim has been taken almost for granted in the paper, while it is not common knowledge and requires justification. We have now revised our discussion in the main paper and added an in-depth review of existing CL algorithms and their link to the MT objective in the appendix (Section B in the new version). In general, CL algorithms do not implement the *exact* MT loss. Instead, they can often be viewed as biased estimates of it, where the MT loss is either weighted differently across tasks or approximated by regularization terms that may deviate from the original objective. We still believe that studying the MT objective offers a high level intuition on many continual learning algorithms. However, we acknowledge that the precise level of 'suboptimality' for specific CL methods is not clear. To determine this, our theoretical framework has to be applied to the specific objective of each method.
>
> 5. Regarding the description of the permutation experiments, it can be found in the appendix (section C.1.4., Figures 8 and 9 in the new version).
>
>
> [1] Soutif–Cormerais, Albin, Antonio Carta, and Joost Van de Weijer. "Improving online continual learning performance and stability with temporal ensembles." Conference on Lifelong Learning Agents. PMLR, 2023.
>
> [2] Cai, Zhipeng, Ozan Sener, and Vladlen Koltun. "Online continual learning with natural distribution shifts: An empirical study with visual data." Proceedings of the IEEE/CVF international conference on computer vision. 2021.

---

> > ### Comment · Reviewer_cvzj · 2024-11-25
> >
> > Thanks to the authors for the nice answers, as well as the nice amount of modifications to the manuscript accounting for the reviewers opinions. I will increase my score to 6 to account for the effort and also to show my agreement to the change in point of view on continual learning that this paper is trying to push.
> >
> > However, I still think more work is required to push in that direction. Although a theoretical approach is interesting I don't think this is the most efficient way to convince people that a change of point of view is required. My intuition is that, even if this change happens (that we care more about learning speed and average lifetime error), having access to all previous data will always be beneficial (But maybe using the multitask objective is not).
> >
> > I think in order to convince the community one would need to refer to metrics that have been previously used like forgetting, and turn them into something new. For instance, showing a comparison of a continual learning method that manages to keep some knowledge of one task (buried inside the network weights) but at the same time lets his feature drift in order to learn new task with high plasticity (so it will have high classical forgetting potentially). Then when revisiting the task, we should see that the CL method is able to adapt quicker to the revisited task compared to the Naive baseline not performing CL. What I mean by this example is that changing point of view does not mean the previous concerns of forgetting disappear, but just that the way we measure it changes (and that measuring speed of adaptation could be another way of measuring forgetting if task revisital is considered), and I don't think this paper does a good job at doing this because it does consider task revisital, which I think is essential to consider in this new point of view.

---

> ### Author Response · Authors · 2024-11-29
>
> We thank the reviewer for acknowledging our efforts and for the stimulating discussion. We would like to continue the conversation regarding some of the ideas raised.
> The reviewer suggests that, to shift the community’s focus toward adaptivity, we should redefine existing metrics such as ‘forgetting’ to align with this focus, for instance, by redefining forgetting in terms of the speed of adaptation when an old task is encountered.
> While we understand the reviewer’s point, we believe that reinterpreting existing metrics could lead to confusion. That said, we fundamentally agree with the reviewer that, in the context of forward transfer, lower forgetting of past tasks correlates with faster adaptation when a task is revisited. Intuitively, minimizing forgetting becomes particularly important when the likelihood of encountering a task again is high. Thus, incorporating assumptions about the probabilistic structure of the environment could help determine the appropriate level of forgetting. This aligns with our proposition that future research in continual learning (CL) should focus on developing data-dependent objectives that are provably optimal.
>
> The goal of this work is to demonstrate that the currently prevalent multitask objective is suboptimal, thereby opening the door for discussions like the one we are having in this rebuttal and encouraging future research on alternative objectives for CL.
> Finally, we would like to note that our theoretical analysis can account for ‘task revisitation’ as it does not make assumptions about task similarity. Specifically, task revisitation would reduce the instability of the task sequence, depending on the ordering. In the toy setting presented in the paper, the same two tasks are revisited alternately in a repetitive manner.

---

### Official Review · Reviewer_JWtx · 2024-11-03

**Soundness:** 3
**Presentation:** 3
**Contribution:** 2
**Rating:** 6
**Confidence:** 3

**Summary:**

This paper studies CL under a new metric borrowed from Online Learning, called dynamic regret. It is shown that under this new metric, matching a multitask learner, which is a common goal in the literature, might be suboptimal with respect to dynamic regret. Specifically, performance of a single task learner is compared  to that of a multi task learner in a linear model to gain insight into when multitask learner is optimal. Empirical analysis supporting the theoretical findings is provided.

**Strengths:**

This paper is studying a question that is important to the community, and is well motivated. The arguments are laid out mostly clearly and are easy to follow. The experiments are extensive.

**Weaknesses:**

I think the main caveat of this work is the underlying assumption that the risk of the defined multi task agent, which sequentially trains on tasks starting from the previous solution, converges (with number of steps $h \rightarrow \infty$) to the risk of true multi-task solution which is a minimizer of average risk of tasks seen so far.  Using the notation in equation (2),  the claim is that $\Delta_T^{MT} \rightarrow 0$. This assumption is not explicit, it is mentioned  in line 196, that it holds in convex settings. Looking in the appendix section B.1.2, however, it seems that MT agent defined in the linear convex setting takes gradient steps with respect to an objective that takes into account all tasks simultaneously. So this does not match the description of the MT agent given in line 169, which I think needs to be clarified.
My understanding is that it is not easy to match the performance of a true multitask learner (that minimizes error on all tasks simultaneously ) while learning continually.
The empirical analysis section is a little bit hard to follow. It is not always easy to follow which part of the narrative each figure/paragraph supports. Some examples:


 - Table 3 : not sure what to expect by looking at the number of tasks. What is the hypothesis here?
 - Table 1: it seems that $v_{agent}$ is tracking error while $O_{agent}$ is tracking accuracy.

Description of the algorithm Selective Replay is missing from the main text.

**Questions:**

It would be great if the authors could explain the discrepancy between the linear MT agent and then one that trains continually on one task at a time.

Suggestions:

- I think moving equation 19 to main text and moving equation 4 would be helpful. Is there a $\Sigma_x$ subscript missing from equation 4?
- Include definition of SR in the main text (I could not find it in the appendix)
- line 457: says instability is higher for  PC-16.

---

> ### Author Response · Authors · 2024-11-19
>
> We would like to thank the reviewer for the thoughtful and constructive feedback on our paper. We appreciate the opportunity to clarify our work and make improvements based on your comments. We would like to point out that we have switched the order of the sections in the Appendix, the theoretical results now come first.
>
> 1. Regarding the main caveat of the work, we believe there is a misunderstanding.  We have modified Section 3 and 4 in order to improve the exposition of our theoretical analysis, and we hope to have resolved this confusion.
>
>    In particular, the MT agent (which can be seen as a 'perfect' replay agent with infinite buffer, representing an agent that minimizes forgetting) is trained on the "average error across all tasks encountered up to the current point" (line 155 in the revised paper), and indeed in the appendix (now A.1.2.) the MT objective during task $\kappa$ is equal to the sum of errors $R_i$ where $i\le \kappa$. Thus, the MT GD agent 'continually descends', meaning that it is not reset at the end of a task, but keeps its previous task estimate as initialization. The assumption which you mentioned in line 196 (line 210 of the new version), specifically $\frac{1}{K} \Delta_T^{MT} \in o(1)$, is proven in the convex setting in Lemma 9 (of the new version). It is a simple consequence of the fact that the contribution of each task to the new average decreases as the task index increases. We also recognize that the previous definition of key quantities in the theoretical sections was sometimes sloppy, so we have thoroughly polished Section 3 and 4. We kindly ask the reviewer to have a look at the new version and share their thoughts.
>
> 2. We significantly strengthened our theoretical analysis, by adding new results and claims. Specifically, we derived a matching lower bound for $\Delta T$, which allowed us to establish tighter bounds, and explicitly describe the dependency on h. Additionally, we expanded our derivations to cover the overparameterized case, and refining the previously inconsistent exposition of theoretical results. We believe these improvements consolidate the theoretical foundation of our work.
>
> 3. The results in Table 3 are intended to evaluate the effect of increasing the number of tasks. From our theoretical result the variable $K$ only indirectly affects the balance between MT and ST objectives, through the instability. Thus, the experiments of Table 3 are meant to evaluate in practice what the effect of $K$ is. We have revised the presentation of Table 3, and, more generally, Section 5 to make the underlying hypotheses more explicit and easier to follow.
>
> 4. We have added a detailed description of the Selective Replay algorithm in the main text to address the concerns about its omission.  This addition should provide better context for understanding the experiments and their outcomes. We thank the reviewer for highlighting this point.

---

> > ### Comment · Reviewer_JWtx · 2024-11-28
> >
> > Thanks for the clarification, this addresses my main concern. One limitation that does remain is that this MT agent is not realistic, and even with some replay, the agents used in practice might still focus on the most recent task mostly. However, as a first step, studying this idealized agent makes sense to me. I do think this should be highlighted in the limitations.  I have increased the score.

---

> > > ### Author Response · Authors · 2024-11-28
> > >
> > > Thank you for the feedback. We have addressed this point in the limitations section.
> > >
> > > We acknowledge that the multitask (MT) agent is 'idealized'—no existing algorithm with constrained computational or memory resources can fully match it. However, we argue that the multitask objective is a valid and useful choice in the context of this work, as it enables a clean and focused analysis. Specifically, this work addresses whether maximizing average performance across tasks, a longstanding goal in continual learning (CL), is beneficial for lifelong online performance. The multitask objective inherently maximizes the average performance across tasks, and as such, the MT agent's average performance serves as an 'upper bound' for the average performance of any practical CL agent. In this sense, the objective of a practical CL agent can be viewed as a potentially biased approximation of the multitask objective.

---

### Official Review · Reviewer_8Lkj · 2024-11-05

**Soundness:** 3
**Presentation:** 2
**Contribution:** 2
**Rating:** 6
**Confidence:** 3

**Summary:**

This paper challenged a long-standing assumption in continual learning (CL): multi-task learning (MTL) is the upper bound for CL. Authors found that MTL is not always the upper bound for CL especially in highly non-stationary environments or long sequences. To explain their findings, theoretical results showed that the single-task system is more suitable in a volatile environment. Experiments were conducted to confirm the hypotheses and theoretical results in both synthesis and real-world environments.

**Strengths:**

This paper has several strengths:

- This paper questions a popular but underexplored assumption in CL: is MTL always an upper bound for CL system? The authors show that this is not true in highly complex environments both theoretically and empirically. Answering this question allows us to understand when we should ignore the MTL results in a benchmark and explain why several CL methods yield better results than MTL.

- The authors conducted a comprehensive experiment to verify their hypothesis. The empirical results verify their theoretical results.

**Weaknesses:**

Despite these strengths, my main concern is about the contribution of this work:

- Several parts of the paper need more clarification for smoother reading and understanding. I struggled during reading Sections 3 and 4 with several notations that were not fully explained. E.g., the $\theta^*$ in Eq.4. Many typos in paragraphs of the main paper such as in line 370. I recommend authors carefully revise the main paper during the rebuttal process.

- Since the DL models are mostly overparameterized, the theoretical results only consider the linear models in a strictly convex setting, limiting the contributions of this work. I wonder what happens if we add the regularization term in the loss function as indicated in the discussion part.

- Although the authors pointed out that there are some cases that the MTL is not as good as ST, I wonder: is there any recommendation, signal, or measure for practitioners to recognize these cases before training and estimate the instability? it would make this work more solid.

- Despite in Sec.3 and Sec.4, the authors emphasize that the setting of this paper is online learning. However, in the experiments, authors use h = 3000, 6000,... In my opinion, it resembles the offline continual learning setting when each task is trained for several epochs not a single one like in online continual learning. Is there any explanation for this?

**Questions:**

See the weaknesses.

---

> ### Author Response · Authors · 2024-11-19
>
> We thank the reviewer for their valuable suggestions, which have significantly helped improve the paper. Below, we address the specific points raised:
>
> 1. We have improved the overall presentation, especially in Sections 1 to 4. We also added the definition of $\theta^\star$ at the beginning of Section 4. We have corrected various typographical errors, including inconsistencies in notation, grammatical issues, and minor formatting mistakes that were present throughout the manuscript.
>
> 2. You correctly noted that our theoretical results are derived under the assumption of linear models in a strictly convex setting. This limitation constrains the generalizability to overparameterized models. We have extended our theoretical analysis to discuss how to handle the overparametrized case in greater detail, which can be found in Section A.4 in the Appendix material. Importantly, the results do not change at a fundamental level.
>
> 3. The question of deciding between objectives is indeed fundamental to continual learning, and our paper emphasizes this. We acknowledge that the current estimates of instability may be impractical due to their computational cost; however, in many cases, a task transfer matrix or equivalent task structure information could suffice. We believe this presents a promising direction for future research. To explore this further we attempted to measure instability at initialization using the Neural Tangent Kernel (NTK). The results are preliminary, and thus we opted not to include them in the main text, but they can be found in Section A.5. We look forward to your feedback on this aspect.
>
>
> 4. Regarding the confusion about the online versus offline learning nature of the training: our setup is indeed atypical, and this is intentional. Our goal is to determine whether the multi-task objective is optimal with respect to a lifelong performance metric, such as the average lifelong error (or dynamic regret). While our evaluation metric is online -meaning that it only considers the current task error- our training paradigm does not need to be. We decouple the training and evaluation protocols, allowing for potentially offline training.
> We have clarified this distinction in the introduction and the background sections (see lines 120-130).

---

> > ### Comment · Reviewer_8Lkj · 2024-11-25
> >
> > I truly appreciate author's efforts to clarify my major my concerns and a large amount of work has been done in such a short time. However, my concern about the online setting is still remaining. In particular, why do we need the evaluation protocal is online while the traing procedure is offline. I hope authors can clarify it. I will update my rating after my concern is fully addressed.

---

> > > ### Author Response · Authors · 2024-11-27
> > >
> > > Thank you for your thoughtful feedback and for acknowledging our efforts to address your concerns. Regarding the online evaluation protocol, we would like to clarify that ''online evaluation'' refers to evaluating on the current task only, rather than on all tasks observed so far. By this definition, the ST agent's training procedure is considered online, as the training objective is computed solely on the current task data. Our goal in this work is to assess whether learning algorithms with memory of past experiences (represented by the MT agent) can perform well in dynamic and interactive scenarios. To this end, we use an online evaluation approach to align with real-world deployment settings, where the model interacts continuously with a changing environment, while keeping the training process offline to leverage past data. This distinction allows us to better evaluate the model's robustness and adaptability in practice.
> > > We hope these explanations align with your expectations. We would be happy to provide further clarifications if needed.

---

> > > > ### Comment · Reviewer_8Lkj · 2024-11-28
> > > >
> > > > Thank you for your feedback. Your rebuttal addressed most of my concerns. I increased the score accordingly.

---

### Author Response · Authors · 2024-11-19
**General comment to all reviewers**

Overall, despite the low scores, the general sentiment of the reviews is constructive. All the reviewers' have raised very valid points. As highlighted by the reviews, our work was lacking in presentation, and we believe that the paper quality has increased dramatically by incorporating the reviewers' feedback. Thus, regardless of the scores, we are very grateful to the reviewers and their work. In the past days, we have put a lot of effort into updating the manuscript and we would kindly ask the reviewers to have a (quick) look at the new version uploaded on openreview.  Where possible, we have highlighted the changes made in blue. In the personalized responses, we will refer to the changes which concern the specific question of the reviewer. However, we recommend a complete read of the new version, as we believe that the overall exposition, flow, coherence, and quality have been significantly improved.

What follows is a list of the **main changes applied**:
1. We have modified the text, especially in Sections 1 to 4, increasing clarity, attending especially to the points of confusion raised by the reviewers.
2.  In particular, we expanded on the motivation behind this work, addressing the questions and doubts of Reviewer Dyh5.
3.   Appreciating the suggestion of Reviewer cvzj, we included a discussion of existing online continual learning metrics and the link between this work and that line of literature, which is definitely relevant to our paper.
4.   Given the doubts raised on the connection between MT and CL objectives - which is essential to this work - we expanded on this background topic in Section 2 in the main paper and with a dedicated section in the Appendix.
5.   We greatly improved the theoretical analysis, overall **adding new results and stronger claims** in Section 4 and the Appendix. In particular, we have worked out a matching lower bound for $\Delta_T$. This allows us to provide tight bounds on $\Delta_T$ in general, and to define more precisely the nature of the dependency on $h$. Moreover, we have expanded our derivations to cover the overparametrized case. In addition to expanding it, we have also cleaned the (previously sloppy) exposition of the theoretical results.
6.   We have improved our toy settings, prompted by Reviewer cvzj (and updated accordingly the math in the appendix) and included simulations to estimate the critical task duration. We believe now the toy settings make a much clearer exemplification of all the factors which we have isolated in the theory.
7.   We have greatly improved the exposition of the experimental section. Following the suggestion of Reviewer cvzj and JWtx, we have added a description of the structure of the experimental section at the beginning, and we have placed the tables close to the respective paragraph. We have also changed all the supervised learning metrics from errors to accuracies to improve the interpretability and align better with the rest of the literature.

We hope that our efforts will be noticed, and that - if the reviewers also find the paper improved - they would promote acceptance to the conference. Especially considering that all the feedback pointed out problems on the presentation we believe this work is not lacking any substantial component and is already complete.

---

> ### Author Response · Authors · 2024-11-24
>
> Dear all, as the rebuttal period nears its conclusion, we kindly remind reviewers to actively participate in the rebuttal discussion. We have implemented substantial revisions to the paper in response to the insightful feedback provided, which we believe has significantly enhanced its quality. We would greatly appreciate it if you could take a moment to review the updated version and share your thoughts by providing a response at your earliest convenience. Thank you for your valuable contributions to this process!

---

### Author Response · Authors · 2024-11-28
**Final revised manuscript**

Dear all, given that the deadline for the revision period is today, we have uploaded a new revision with minor changes:
- we have moved the discussion of limitations to a separate section in the Appendix, in response to the concerns of Reviewer Dyh5
- we have included in the limitations some considerations regarding the MT objective,  as suggested by Reviewer JWtx

We thank all the reviewers for the input.

---

### Meta-Review · Area_Chair_SfHu · 2024-12-20

**Metareview:**

This paper studies the problem of episodic continual learning, in which a neural network aims to learn new predictive tasks throughout learning without forgetting original tasks. The primary contribution of this paper is theoretically and empirically demonstrating that the multitask learning setup - which is often assumed to be "an upper-bound of what the learning process can achieve" - can be surpassed by a single-task learner under the online learning metric of dynamic regret.

In many ways, the paper is well-executed: it is written clearly, contains sound theory and experiments backing up the theory, and challenges a long-held assumption of continual learning. Nevertheless, this paper could benefit from an additional round of reviewing to address remaining concerns about significance and scope. The new metric introduced for continual learning can be perceived as being biased against multitask learning, so the paper would benefit from a comparison to other metrics more commonly used by the community. Moreover, the primary finding - as it is presented in the paper - can be interpreted as a simple refinement of the well-established observation that multitask learning is suboptimal when tasks are not sufficiently related.

Shifting the focus of the claims will significantly strengthen this paper, especially since the claims revolve around "a topic of ongoing debate within the CL community."

**Additional Comments On Reviewer Discussion:**

The discussion between the authors and reviewers was polite, thorough, and constructive. The reviewers had minor technical concerns (e.g. simplicity of toy setting, questions about experimental procedure, etc.), though all reviewers agreed that the authors sufficiently addressed these concerns in the revision. The primary discussion around this paper focused on the significance of the claims, which were primarily up for debate because the authors introduced a new metric to demonstrate the non-optimality of multitask learning. (See above for a summary of the reviewers' concerns.) While the discussion between reviewers and authors was constructive and led to a refinement of claims to address this issue, the reviewers were not sufficiently convinced by the significance of the updated claims. Nevertheless, most reviewers gave actionable suggestions for improving the claims (see above for summary).

---

### Decision · Program_Chairs · 2025-01-22

Reject